# Membrane recruitment of the polarity protein Scribble by the cell adhesion receptor TMIGD1

Eva-Maria Thüring [1,6], Christian Hartmann[1,6], Janesha C. Maddumage [2,6], Airah Javorsky[2], Birgitta E. Michels [1], Volker Gerke [3], Lawrence Banks[4], Patrick O. Humbert [2], Marc Kvansakul [2✉] & Klaus Ebnet [1,5✉]

Scribble (Scrib) is a multidomain polarity protein and member of the leucine-rich repeat and PDZ domain (LAP) protein family. A loss of Scrib expression is associated with disturbed apical-basal polarity and tumor formation. The tumor-suppressive activity of Scrib correlates with its membrane localization. Despite the identification of numerous Scrib-interacting proteins, the mechanisms regulating its membrane recruitment are not fully understood. Here, we identify the cell adhesion receptor TMIGD1 as a membrane anchor of Scrib. TMIGD1 directly interacts with Scrib through a PDZ domain-mediated interaction and recruits Scrib to the lateral membrane domain in epithelial cells. We characterize the association of TMIGD1 with each Scrib PDZ domain and describe the crystal structure of the TMIGD1 C-terminal peptide complexed with PDZ domain 1 of Scrib. Our findings describe a mechanism of Scrib membrane localization and contribute to the understanding of the tumor-suppressive activity of Scrib.

[1] Institute-associated Research Group "Cell adhesion and cell polarity", Institute of Medical Biochemistry, ZMBE, University of Münster, Münster, Germany. [2] Department of Biochemistry & Chemistry, La Trobe Institute for Molecular Science, La Trobe University, Melbourne, VIC, Australia. [3] Institute of Medical Biochemistry, ZMBE, University of Münster, Münster, Germany. [4] International Centre for Genetic Engineering and Biotechnology, Trieste, Italy. [5] Cells-in-Motion Interfaculty Center, University of Münster, Münster, Germany. [6] These authors contributed equally: Eva-Maria Thüring, Christian Hartmann, Janesha C. Maddumage. ✉email: m.kvansakul@latrobe.edu.au; ebnetk@uni-muenster.de

The apical-basal membrane polarity of epithelial cells is determined by three distinct membrane domains: the apical domain facing the lumen of an organ or the outside of a tissue, the lateral domain contacting adjacent cells, and the basal domain adhering to the extracellular matrix (ECM)[1]. The three domains differ in the composition of proteins and lipids to achieve their specific functions, including the formation of a brush border in the apical membrane domain, the regulation of cell-cell cohesion at the lateral membrane domain, and the regulation of cell-matrix adhesion at the basal membrane domain. The development and maintenance of apical-basal polarity is critical, as a loss of cell-cell or cell-matrix adhesion contributes to uncontrolled growth and, eventually, to metastasis, unless counter-regulated by cell death mechanisms such as anoikis[2]. Importantly, the establishment of membrane polarity is a dynamic process. For example, during development or wound healing some epithelial cells lose apical-basal membrane polarity, adopt mesenchymal characteristics, and migrate to distant sites, a process called epithelial-to-mesenchymal transition (EMT)[3]. Conversely, mesenchymal cells can re-polarize at distant sites to form new epithelial tissues[4,5]. Given that transitions between epithelial and mesenchymal states are critical for the induction of normal and also of neoplastic stem cells from somatic cells[5,6], the development of apical-basal polarity must be tightly controlled to prevent neoplastic transformation.

Epithelial apical-basal polarity is regulated by a conserved set of polarity proteins which are organized around three major cell polarity protein complexes. These include the Crumbs – Pals1 – PATJ complex (Crumbs complex), the PAR-3 – aPKC – PAR-6 complex (PAR – aPKC complex), and the Scribble – Dlg – Lgl complex (Scrib complex)[1,7]. During the development of apical-basal polarity, the composition of these three polarity complexes is dynamically regulated and involves mutual interactions and changes in their composition[1,8]. In fully polarized epithelial cells, the Crumbs complex together with a PAR-6 – aPKC complex localizes to the apical membrane. PAR-3 is localized at the tight junction, which demarcates the border between apical and lateral membrane domains, and the Scrib complex is localized at the lateral membrane domain[1,9]. The mutually exclusive localization of the polarity proteins is in part regulated by antagonistic interactions. For example, phosphorylation of PAR-3 by the PAR-6 – aPKC complex prevents the formation of a PAR-3 – aPKC – PAR-6 complex at the apical membrane[10,11]. Also, the polarity kinase PAR-1 inactivates PAR-3 to prevent the formation of a stable PAR-3 – aPKC – PAR-6 complex at the lateral membrane[12].

Given the critical role of epithelial apical-basal polarity in the maintenance of tissue homeostasis it is not surprising that polarity signaling is often dysregulated in cancer[13,14]. Among the polarity proteins involved in tumorigenesis, Scrib stands out as its downregulation has been reported in a large variety of tumors[15,16]. Its pleiotropic functions in suppressing tumor formation can partly be explained by its ability to interact with a plethora of binding partners[15,17] through which Scrib impacts on several signaling pathways involved in cell proliferation and cell migration. These pathways include Hippo signaling, RAS – MEK – ERK signaling, JNK – p38 signaling, and PI3K – Akt signaling[15]. Importantly, mislocalization transforms Scrib from a membrane-associated tumor suppressor to a cytosolic driver of tumor formation[18,19], suggesting that membrane localization is key to its tumor-suppressing activity. Despite this critical role of membrane localization, the mechanisms of membrane recruitment are not fully understood[17]. Scrib comprises of 16 Leucine-Rich Repeats (LRR), two LRR and PSD95/Dlg/ZO-1 (PDZ)-specific domains (LAPSD), and four PDZ domains[16]. While a few interactions are mediated by the LRR region and the C-terminal region, the vast majority of interactions of Scrib are mediated via the PDZ domains, typically by binding a PDZ domain-binding motif (PBM) found at the extreme C-terminus of an interacting protein to the conserved ligand binding groove in the PDZ domain[15,17].

Cell adhesion receptors are frequently involved in cell polarity by recruiting polarity proteins to sites of cell-cell adhesion. For example, junctional adhesion molecules (JAMs), nectins and VE-cadherin interact with cell polarity protein PAR-3 through their C-terminal PBM[20]. Recently, a cell adhesion receptor with similarity to JAMs called Transmembrane and Immunoglobulin Domain-containing protein 1 (TMIGD1) has been identified[21]. TMIGD1 contains two immunoglobulin (Ig)-like domains and a short cytoplasmic tail with a C-terminal PBM[22]. Its expression is successively downregulated during the progression of normal colonic tissue to colorectal cancer tissue[21,23]. TMIGD1 is predominantly expressed by proximal tubule kidney epithelial cells[23–25] and by intestinal epithelial cells[26–28]. In kidney epithelial cells TMIGD1 is localized at mitochondria when cells are grown at low density but redistributes to cell-cell contacts when cells reach confluency[25]. In intestinal epithelial cells TMIGD1 localizes to the brush border and regulates microvilli organization[26,28]. In almost all other tissues TMIGD1 mRNA or protein expression is hardly detectable (https://www.proteinatlas.org/ENSG00000182271-TMIGD1/tissue)[24]. Also, in many cultured cell lines derived from different tissues, TMIGD1 expression is very low[21].

In this study, we identify TMIGD1 as a binding partner for Scrib. We find that TMIGD1 interacts directly with Scrib in a manner that involves PDZ domains of Scrib and the PBM of TMIGD1. We describe the crystal structure of the Scrib PDZ1 domain complexed with a C-terminal TMIGD1 PBM peptide. We show that TMIGD1 can recruit Scrib to the lateral membrane domain and that expression of a dominant-negative TMIGD1 mutant disturbs three-dimensional cyst morphogenesis. Our findings identify TMIGD1 as an adhesion receptor at the lateral membrane of polarized epithelial cells which directly binds Scrib. They also provide a structural basis for the mechanisms regulating Scrib membrane recruitment in epithelial cells.

## Results

**Scribble interacts with the cell adhesion receptor TMIGD1**. In a search for cytoplasmic binding partners of TMIGD1 we have performed yeast-two hybrid (Y2H) experiments using the cytoplasmic domain of TMIGD1 as bait. In these experiments we have previously identified Synaptojanin 2-binding protein (SYNJ2BP) and $Na^+/H^+$ exchange regulatory cofactor 2 (NHERF2) as proteins which directly interact with TMIGD1 in a PDZ domain-dependent manner[22,25,28]. In the same Y2H experiments we have isolated several overlapping clones which covered a total region encoding AA633 to AA836 of murine Scrib (Fig. 1a), suggesting that TMIGD1 interacts with the first PDZ domain of Scrib. GST-pulldown experiments with the cytoplasmic tail of TMIGD1 fused to GST (GST-TMIGD1/f.l.) and a recombinant Scrib construct containing the PDZ domain region of Scrib (Scrib/PDZ1-4) confirmed a strong interaction of TMIGD1 with the Scrib PDZ domains which was lost after deletion of the C-terminal PBM (-SETAL$_{COOH}$) of TMIGD1 (GST-TMIGD1/Δ5, Fig. 1b). These observations identified Scrib as a binding partner of TMIGD1, which like SYNJ2B and NHERF2 directly interacted with TMIGD1 through a PDZ domain-mediated interaction.

In many cultured polarized epithelial cells including Caco2 cells, T84 cells and MDCK cells, TMIGD1 is expressed by only a small subset of cells or not at all[28]. We therefore expressed TMIGD1 ectopically in HEK293 cells and analysed its interaction

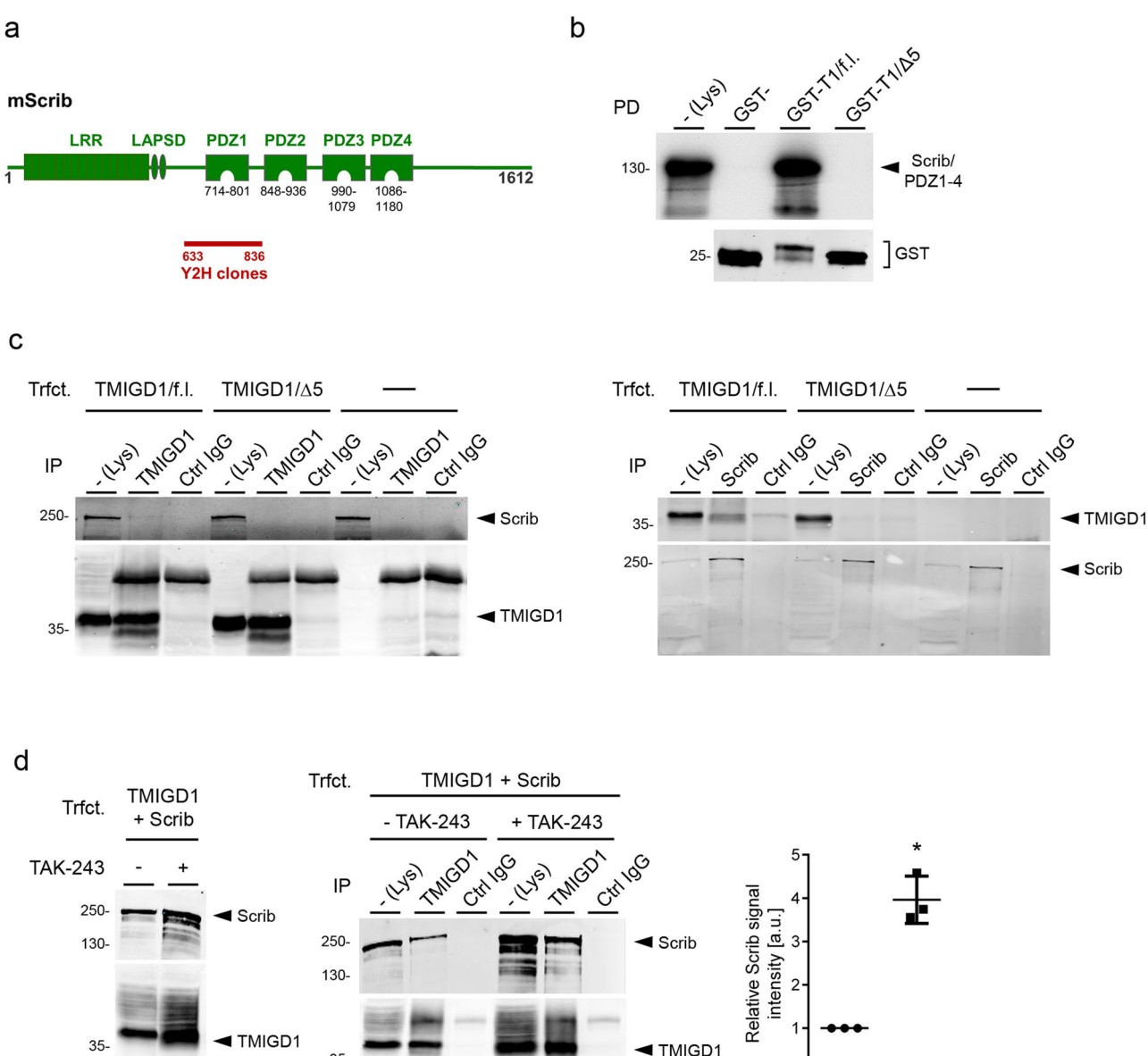

**Fig. 1 Scrib interacts with TMIGD1. a** Schematic organization of murine Scrib. The region isolated in several independent cDNA clones from a Y2H library that interacted with TMIGD1 (AA 633–836) is depicted by a red bar. Nomenclature refers to isoform 1 of murine Scrib. LAP leucine-rich repeat and PDZ domains, LAPSD LAP-specific domain, LRR leucine-rich repeat, Y2H yeast-two hybrid. **b** GST pulldown. GST-TMIGD1 fusion proteins containing the entire cytoplasmic domain of TMIGD1 (GST-T1) or a deletion mutant lacking the PBM (GST-T1/Δ5) were immobilized and incubated with a Scrib construct consisting of the four PDZ domains (Scrib/PDZ1-4) generated in vitro by a coupled transcription/translation system in the presence of $^{35}$S-methionine. Bound proteins were analyzed by autoradiography. GST fusion proteins used in the pulldown experiments were visualized with anti-GST antibodies by Western blotting (GST, 10% of input). PD pulldown. **c** CoIPs from HEK293 cells stably transfected with TMIGD1 constructs (TMIGD1/f.l., TMIGD1/Δ5) or untransfected HEK293 cells. IPs were performed with antibodies against TMIGD1 or with antibodies against endogenous Scrib as indicated. **d** Interaction of TMIGD1 and Scrib in the presence of TAK-243. Left panel: Western blot analysis of lysates of TMIGD1- and Scrib-transfected HEK293 cells in the absence and presence of TAK-243. Middle panel: CoIP of Scrib and TMIGD1. IPs were performed with anti-TMIGD1 antibodies, immunoprecipitates were blotted with antibodies against Scrib or against TMIGD1 (10% input), as indicated. Right panel: Quantification of Scrib amounts co-immunoprecipitated with TMIGD1 in the absence and presence of TAK-243. Signal intensities obtained from mock-treated cells (-TAK-243) were arbitrarily set as 1. Statistical analyses were performed with unpaired Student's *t* test. Data are presented as mean values ± SD. *$P < 0.05$. Ctrl control, Lys lysate, IP immunoprecipitation, TMIGD1/f.l. TMIGD1/full length, TMIGD1/Δ5 TMIGD1 lacking the five C-terminal AA, Trfct. transfection. Data shown in this figure are representative of at least 3 independent experiments.

with endogenous Scrib by co-immunoprecipitation (CoIP). Scrib was present in TMIGD1 full length (TMIGD1/f.l.) precipitates but not in TMIGD1/Δ5 precipitates, and vice versa, TMIGD1/f.l., but not TMIGD1/Δ5, was present in Scrib precipitates (Fig. 1c), indicating that TMIGD1 interacts with Scrib in a PBM-dependent manner in cells. Since this interaction was weak, despite the strong interaction observed with recombinant proteins (Fig. 1b), we ectopically expressed both TMIGD1 and Scrib in HEK293 cells. In addition, since both TMIGD1 and Scrib were described to be subject to proteasomal degradation through the ubiquitin-proteasome pathway[23,29], we performed these experiments in the presence of TAK-243, an inhibitor of Ubiquitin-activating enzyme E1 (UBE1), the mammalian E1 enzyme that charges the vast majority of E2 ubiquitin-conjugation enzymes with ubiquitin[30]. TAK-243 treatment increased the expression levels of both TMIGD1 and Scrib and enhanced the interaction of Scrib with TMIGD1 (Fig. 1d). These findings indicated that both TMIGD1 and Scrib are subject to proteasomal degradation in HEK293 cells, providing a possible explanation why their interaction in cells is difficult to detect.

To further characterize the interaction between TMIGD1 and Scrib we performed CoIP experiments from HEK293 cells expressing various Scrib deletion constructs including Scrib full length (Scrib/f.l.), a construct comprising the four PDZ domains (Scrib/PDZ), a Scrib/PDZ construct with a C-terminal CAAX motif (C, Cys, A, aliphatic AA, X, any AA) to target this construct to endomembranes[31] (Scrib/PDZ-CAAX), and a construct comprising the LRR region of Scrib (Scrib/LRR) (Fig. 2a). The Scrib/f.l. as well as the Scrib/PDZ and Scrib/PDZ-CAAX constructs interacted with TMIGD1/f.l. but not TMIGD1/Δ5 (Fig. 2b, c). The Scrib/LRR construct did not interact with TMIGD1 (Fig. 2d). These findings indicated that TMIGD1 and Scrib interact in cells in a manner that involves the PBM of TMIGD1 and the PDZ domain region of Scrib.

Scrib is a member of the LRR and PDZ (LAP) protein family[32,33] which in mammals consists of Scrib, Erbin, Lano/LRRC1 and Densin-180/LRRC7[16]. Among these, Scrib, Erbin and Lano/LRRC1 are co-expressed in polarized epithelial cells and colocalize at the basolateral membrane domain[34]. Only their combined depletion results in a disorganization of cell-cell junctions indicating a redundancy in their functions[34]. To address the specificity of the TMIGD1 interaction with Scrib, we analyzed the interaction of TMIGD1 with Erbin and Lano/LRRC1 (Fig. 2e). We did not observe an association between TMIGD1 and Erbin or Lano/LRRC1 by CoIP suggesting that TMIGD1 interacts specifically with Scrib among the LAP family members expressed in epithelial cells.

**Scribble PDZ1 interacts with TMIGD1 through a canonical PDZ interaction.** The Y2H screening result as well as the biochemical experiments suggested that Scrib and TMIGD1 interact directly through the Scrib PDZ1 domain and the PBM of TMIGD1. We examined the structural basis of this putative binding interface by determining the crystal structure of the isolated Scrib PDZ1 domain in complex with an 8-mer peptide representing the C-terminal region of TMIGD1 which includes the PBM (-DPHSETAL$_{COOH}$, PBM underlined). The PDZ1:TMIGD1 structure was refined to a resolution of 1.9 Å with a final $R_{free}$ of 0.222 (Supplementary Table 1). As shown previously, the Scrib PDZ1 domain adopts a compact globular fold comprising six β-strands and two α-helices that adopt a β-sandwich structure[35], a conformation that is typical for PDZ domains[36]. The TMIGD1 peptide engaged with the PDZ1 domain via the well-conserved canonical ligand binding groove located between the β2 strand and the α2 helix (Fig. 3a)[35,37–40].

Binding of TMIGD1 to Scrib PDZ1 does not alter the overall domain configuration of Scrib PDZ1 as a superimposition of ligand-free Scrib PDZ1 with the Scrib PDZ1:TMIGD1 yielded a root-mean-square deviation of 0.44 Å, with differences between the two structures primarily due to changes in local side chain conformations.

Close examination of Scrib PDZ1:TMIGD1 peptide complex revealed several direct interactions that are formed between the peptide and the PDZ1 domain (Fig. 3b, Supplementary Fig. 1). The C-terminal carboxyl group of the TMIGD1 peptide docks in the conserved hydrophobic pocket of PDZ1, interacting with the main chain L738$^{PDZ1}$, G739$^{PDZ1}$ and I740$^{PDZ1}$. Furthermore, a network of hydrogen bonds is observed between PDZ1 and TMIGD1 including interactions mediated by PDZ domain side chains (H793$^{PDZ1}$:T260$^{TMIGD1}$, S748$^{PDZ1}$:S258$^{TMIGD1}$, T749$^{PDZ1}$:D255$^{TMIGD1}$) and interactions mediated by the PDZ domain main chain (I742$^{PDZ1}$:A261$^{TMIGD1}$, I742$^{PDZ1}$:T260$^{TMIGD1}$, G744$^{PDZ1}$:S258$^{TMIGD1}$, T749$^{PDZ1}$:P256$^{TMIGD1}$). Lastly, a salt bridge is found between R762$^{PDZ1}$ and E259$^{TMIGD1}$ (Table 1).

A comparison of Scrib PDZ1 complexes with known PBM ligands reveals that key interactions are conserved in the Scrib/PDZ1 – TMIGD1 PBM complex (Table 1). All hydrogen bonds observed in previously characterized Scrib:PBM interactions are recapitulated in the Scrib/PDZ1 – TMIGD1 PBM structure. Consequently, the Scrib/PDZ1 – TMIGD1 is only unique in the sense that a different subset of known PDZ1 – PBM interactions is used to achieve binding.

**Scribble interacts with TMIGD1 through PDZ1 and PDZ3 domains.** Given the extensive interaction network of Scrib[15,17,33] we aimed to characterize the Scrib – TMIGD1 interaction in more detail. To address the possibility that Scrib interacts with TMIGD1 through additional PDZ domains, we performed in vitro binding assays using the cytoplasmic domain of TMIGD1 fused to GST and in vitro translated Scrib PDZ domain constructs in which individual PDZ domains were inactivated by mutating the GLGF motif[41] (Fig. 4a). Mutating PDZ1 strongly reduced binding to TMIGD1, as expected (Fig. 4b). Mutating PDZ2 and PDZ3 also strongly reduced the binding to TMIGD1 suggesting that PDZ2 and PDZ3 are also involved in the interaction with TMIGD1. Mutating PDZ4 did not reduce the binding to TMIGD1 (Fig. 4b). These observations thus indicated that Scrib interacts with TMIGD1 not only through PDZ1 but also through PDZ2 and PDZ3. Direct interactions of Scrib with the same ligand through several of its PDZ domains have been observed for many other Scrib ligands[15,17], which highlights Scrib's role as a multifunctional scaffold for the assembly of diverse multiprotein complexes.

Overlapping preferences of the Scrib PDZ domains against the same ligand can be fine-tuned by different mechanisms including differential affinities[15]. To understand how PDZ1, PDZ2 and PDZ3 of Scrib contribute to the interaction with TMIGD1 we examined the affinities of the 8-mer peptide representing the TMIGD1 C-terminus towards the individual recombinant PDZ domains by isothermal titration calorimetry (ITC) (Fig. 4c). A similar peptide with mutations at positions 0 and -2, which classify the TMIGD1 PBM as Class I[42] (-DPHSEAAA$_{COOH}$), was used as control. We observed affinities of 18.17 µM and 9.12 µM for PDZ1 and PDZ3, respectively (Table 2, Supplementary Table 2), which are in a similar range as observed for other Scrib PDZ1 and PDZ3 ligands including APC[38], β-PIX[35], MCC[37], Human T-cell leukemia virus 1 (HTLV-1) protein Tax1[39], and Vangl2[40] (Table 3). These findings indicated that both PDZ1 and PDZ3 autonomously bind TMIGD1, and that TMIGD1 binds to Scrib PDZ3 with a

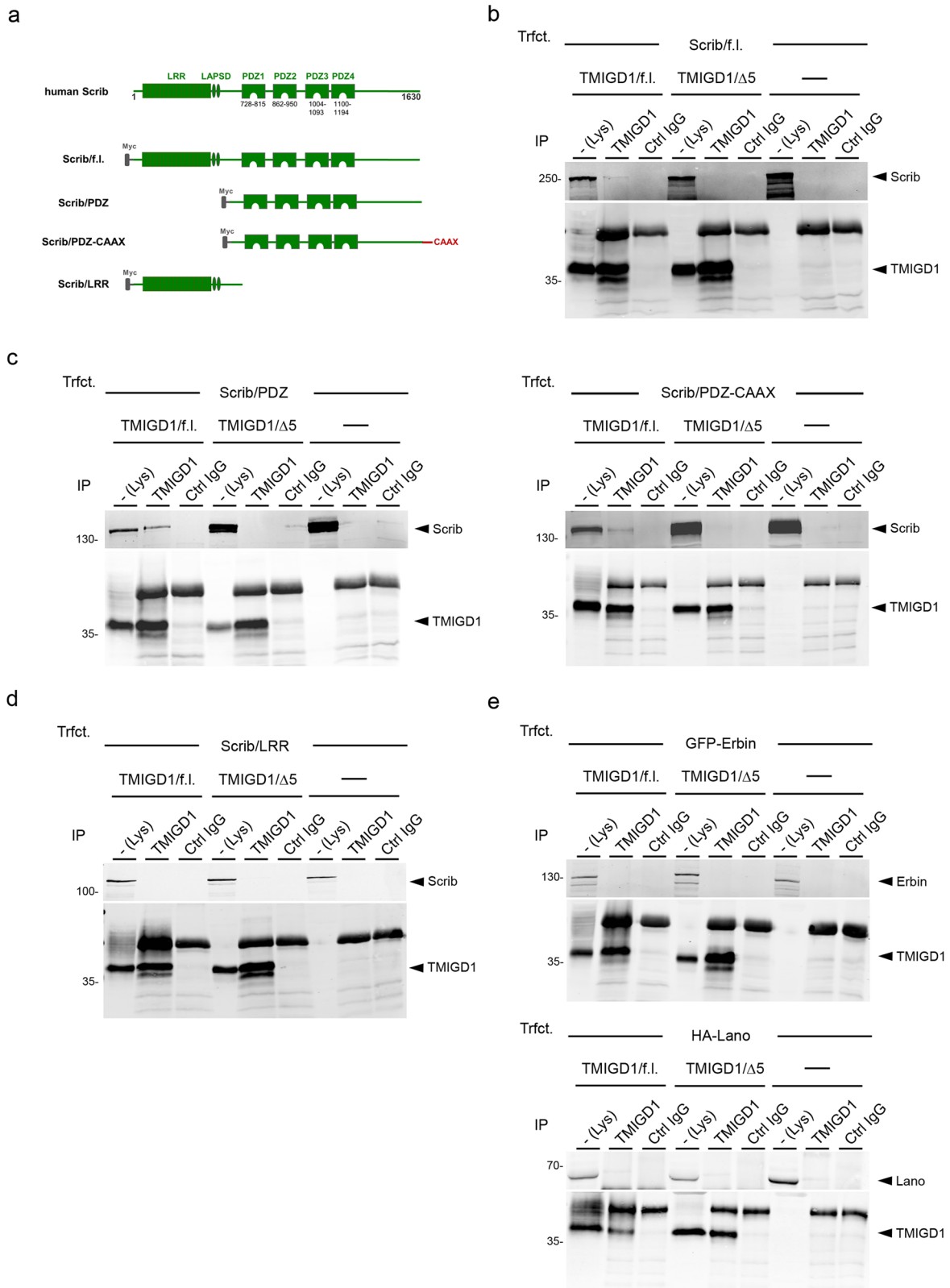

**Fig. 2 Scrib interacts with TMIGD1 through its PDZ domain region. a** Schematic organization of human Scrib constructs used for mapping experiments. **b–d** CoIP experiments with HEK293 cells stably transfected with TMIGD1 constructs (TMIGD1/f.l., TMIGD1/Δ5) or untransfected HEK293 cells and transiently transfected with Scrib constructs shown in Fig. 2a. In all cases, IPs were performed with anti-TMIGD1 antibodies, immunoprecipitates were blotted with antibodies against Scrib or against TMIGD1 (10% input) as indicated. **e** CoIP experiments with HEK293 cells stably transfected with TMIGD1 constructs (TMIGD1/f.l., TMIGD1/Δ5) or untransfected HEK293 cells and transiently transfected with Erbin or Lano/LRRC1. IPs were performed with anti-TMIGD1 antibodies, immunoprecipitates were blotted with antibodies against GFP-Erbin (anti-GFP) and TMIGD1 (top) or against HA-Lano/LRRC1 (anti-HA) and TMIGD1 (bottom). Lys lysate, IP immunoprecipitation, Trfct. transfection. Data shown in this figure are representative of 3 independent experiments.

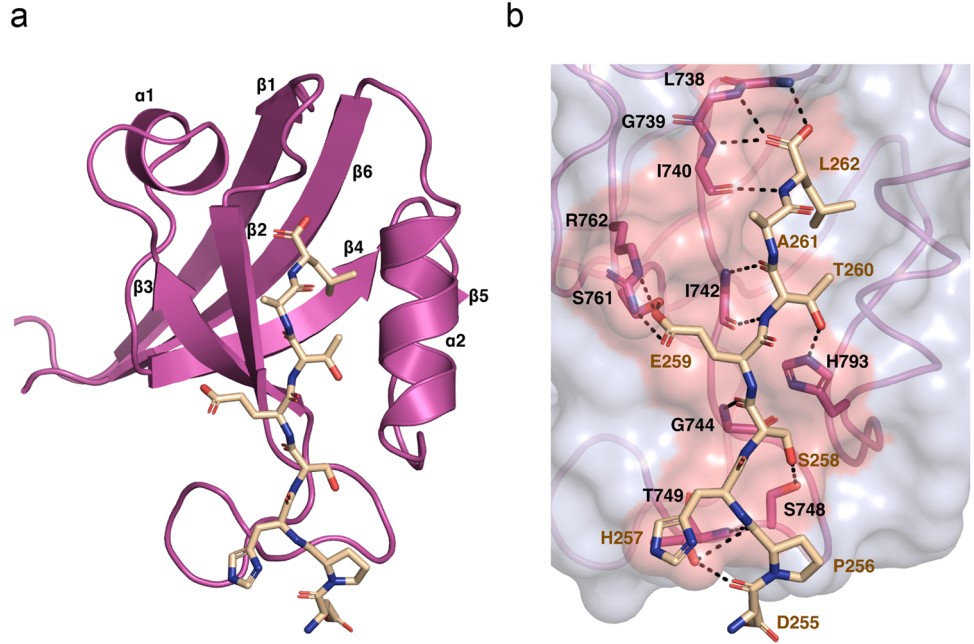

**Fig. 3 Crystal structure of the human Scrib/PDZ1 – human TMIGD1 PBM complex. a** Human Scrib/PDZ1 (light magenta, represented as simplified ribbon diagram, "cartoon") in complex with the human TMIGD1 C-terminal peptide (DPHSETAL, cream, represented as sticks). **b** Detailed view of the Scrib/PDZ1—TMIGD1 interface. The PDZ1 surface and the ligand binding groove are shown in light gray and salmon, respectively; the PDZ1 carbon backbone is depicted as sticks in light magenta. The TMIGD1 peptide is depicted as sticks in cream, oxygen atoms are red, nitrogen atoms are blue. Direct contacts between TMIGD1 peptide and Scrib/PDZ1 protein residues are displayed as dashed black lines. AA residues are displayed in single letter code.

**Table 1 TMIGD1 AA residues involved in direct contacts with Scrib/PDZ1 in comparison with other Scrib-interacting proteins.**

| Pos. C-term. | TMIGD1 | Scrib/PDZ1 | APC | Scrib/PDZ1 | β-PIX | Scrib/PDZ1 | Vangl2 | Scrib/PDZ1 |
|---|---|---|---|---|---|---|---|---|
| -7 | $D_{255}$ --- | --- $T_{749}$ | $G_{2836}$ | – | $P_{639}$ | – | $R_{514}$ | – |
| -6 | $P_{256}$ ••• | ••• $T_{749}$ | $S_{2837}$ --- | --- $T_{749}$ | $A_{640}$ ••• | ••• $S_{748}$ | $L_{515}$ | – |
|  |  |  | $S_{2837}$ ••• | ••• $T_{749}$ | $A_{640}$ ••• | ••• $S_{748}$ |  |  |
| -5 | $H_{257}$ | – | $Y_{2838}$ | – | $W_{641}$ | – | $Q_{516}$ --- | --- $T_{749}$ |
| -4 | $S_{258}$ --- | --- $S_{748}$ | $L_{2839}$ | – | $D_{642}$ ••• | ••• $G_{744}$ | $S_{517}$ | – |
|  | $S_{258}$ ••• | ••• $G_{744}$ |  |  |  |  |  |  |
| -3 | $E_{259}$ ::: | ::: $R_{762}$ | $V_{2840}$ | – | $E_{643}$ --- | --- $S_{761}$ | $E_{518}$ --- | --- $S_{761}$ |
|  |  |  |  |  |  |  | $E_{518}$ ::: | ::: $R_{762}$ |
| -2 | $T_{260}$ ••• | ••• $I_{742}$ | $T_{2841}$ --- | --- $H_{793}$ | $T_{644}$ --- | --- $H_{793}$ | $T_{519}$ ••• | ••• $I_{742}$ |
|  | $T_{260}$ --- | --- $H_{793}$ | $T_{2841}$ ••• | ••• $I_{742}$ | $T_{644}$ ••• | ••• $I_{742}$ | $T_{519}$ --- | --- $H_{793}$ |
| -1 | $A_{261}$ ••• | ••• $I_{742}$ | $S_{2842}$ | – | $N_{645}$ --- | --- $S_{741}$ | $S_{520}$ | – |
| 0 | $L_{262}$ ••• | ••• $L_{738}$ | $V_{2843}$ --- | --- $R_{801}$ | $L_{646}$ ••• | ••• $L_{738}$ | $V_{521}$ ••• | ••• $L_{738}$ |
|  | $L_{262}$ ••• | ••• $G_{739}$ | $V_{2843}$ ••• | ••• $L_{738}$ | $L_{646}$ ••• | ••• $G_{739}$ | $V_{521}$ ••• | ••• $G_{739}$ |
|  | $L_{262}$ ••• | ••• $I_{740}$ | $V_{2843}$ ••• | ••• $G_{739}$ | $L_{646}$ ••• | ••• $I_{740}$ | $V_{521}$ ••• | ••• $I_{740}$ |
|  |  |  | $V_{2843}$ ••• | ••• $I_{740}$ |  |  |  |  |

*APC* adenomatous polyposis coli, *β-PIX* PAK-interacting exchange factor beta, *C-term.* C-terminus, *Pos.* position, *Vangl2* Vang-like protein 2.
Hydrogen bonds mediated by PDZ domain side chains are depicted by broken lines (---), hydrogen bonds mediated by PDZ domain main chains are shown in dotted lines (•••), salt bridges are depicted by colons (:::). The eight C-terminal AA residues of each Scrib/PDZ1 ligand are shown, their positions relative to the carboxy-terminal end (position 0) are depicted in the left column. Note that hydrophobic interactions mediated by the carboxy-terminal AA of the ligands (positions 0) are not shown.

two-fold higher affinity than it binds PDZ1. Surprisingly, despite the strong reduction in TMIGD1 binding affinity when PDZ2 is mutated (Scrib/PDZ2-mut, Fig. 4b), the recombinant PDZ2 alone did not bind TMIGD1 (Fig. 4c, Supplementary Table 2). PDZ4 did not show any affinity towards TMIGD1 (Fig. 4c, Table 2). None of the four Scrib PDZ domains interacted with the control peptide (Table 2). These observations identified PDZ1 and PDZ3 as the PDZ domains of Scrib that directly interact with TMIGD1, and they established a hierarchy between these two PDZ domains. The finding that the isolated PDZ2 does not bind to TMIGD1 whereas the PDZ2 mutant in the context of the Scrib/PDZ domain construct impairs binding to

TMIGD1 opens the possibility that regions adjacent to PDZ2 may influence ligand interactions with PDZ2[43,44].

**TMIGD1 recruits Scribble to cell-cell contacts.** Our observations of a direct PDZ domain-mediated interaction of Scrib with TMIGD1 identified TMIGD1 as an epithelial cell-cell adhesion receptor which directly interacts with Scrib. To test if TMIGD1 recruits Scrib to the basolateral membrane, we ectopically expressed myc-tagged Scrib constructs (depicted in Fig. 2a) in HEK293 cells stably transfected with TMIGD1/f.l. or TMIGD1/Δ5. We first analyzed the localization of endogenous Scrib in TMIGD1-transfected cells by confocal microscopy and found

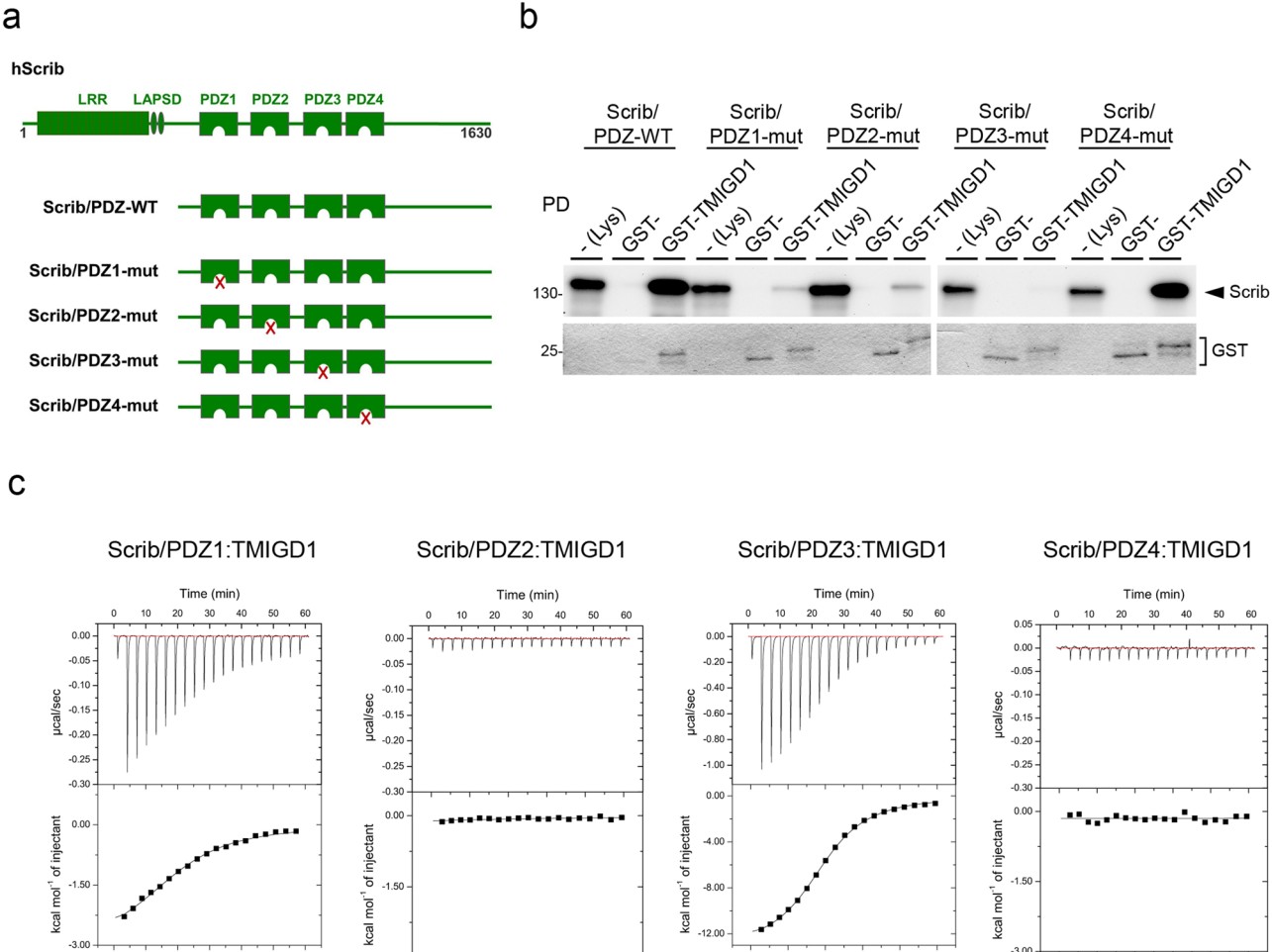

**Fig. 4 TMIGD1 interacts with Scrib through several PDZ domains. a** Schematic organization of human Scrib PDZ domain constructs with inactivated PDZ domains. Mutated PDZ domains are indicated by red crosses. **b** GST pulldown. GST-TMIGD1 fusion proteins were immobilized and incubated with recombinant Scrib proteins with inactivated PDZ domains generated in vitro by a coupled transcription/translation system in the presence of [35]S-methionine. Bound proteins were analyzed by autoradiography. Loading control of GST fusion proteins used in the pulldown experiments were visualized by staining with Coomassie Brilliant Blue (GST, 10% of input). GST-T1 GST-TMIGD1, PD pulldown. Data are representative of 3 independent experiments. **c** Interaction profiles of hScrib PDZ domains with hTMIGD1 peptide. Each profile is depicted by a raw thermogram (top part) and binding isotherm fitted with a one-site binding mode (bottom). $K_D$ dissociation constant ± standard deviation (SD); NB no binding. Each value was calculated from at least three independent experiments.

| Table 2 Comparison of ITC data obtained from WT and mutant TMIGD1 peptides. | | | | | |
|---|---|---|---|---|---|
| **TMIGD1 peptides** | **Sequence** | **Scrib-PDZ1** | **Scrib-PDZ2** | **Scrib-PDZ3** | **Scrib-PDZ4** |
| TMIGD1/WT | DPHSETAL$_{COOH}$ | 18.17 ± 1.80 μM | NB | 9.12 ± 0.67 μM | NB |
| TMIGD1/mut | DPHSEAAA$_{COOH}$ | NB | NB | NB | NB |
| Each of the values was calculated from at least three independent experiments. | | | | | |

that endogenous Scrib co-localized with TMIGD1 at the basolateral membrane domain of HEK293 cells (Supplementary Fig. 2). Ectopically expressed Scrib full length localized to cell-cell contacts irrespective of the presence of TMIGD1/f.l. or TMIGD1/Δ5 (Fig. 5a), suggesting TMIGD1-independent recruitment mechanisms for Scrib, which is consistent with observations of multiple potential binding partners for Scrib[17]. The Scrib/LRR construct was exclusively localized in the cytoplasm (Fig. 5a), indicating that PDZ domains are necessary for membrane localization of Scrib. The Scrib/PDZ construct was localized in the cytoplasm of wildtype HEK293 cells, but was

recruited to cell-cell contacts in TMIGD1/f.l.-expressing cells in a manner that depended on the PBM of TMIGD1 (Fig. 5a). These findings indicated that in the absence of the LRR region, which contains the Pro305 residue critical for membrane localization[18], TMIGD1 recruits Scrib to cell-cell junctions through a PDZ domain-dependent interaction. We also analyzed the recruitment of Scrib/PDZ constructs with mutations in PDZ domains 1, 2 and 3. All three mutant constructs were efficiently recruited by TMIGD1 to cell-cell contacts (Supplementary Fig. 3) indicating that multiple PDZ domains can mediate Scrib membrane recruitment in cells.

**Table 3 Binding affinities of the TMIGD1 8-mer peptide to Scrib PDZ1 and PDZ3 in comparison with the affinities of 8-mer peptides derived from other Scrib-interacting proteins including APC, β-PIX, MCC, HTLV-1 protein Tax1, and Vangl2.**

| Scrib ligand | PBM peptide sequence | Scrib-PDZ1 | Scrib-PDZ2 | Scrib-PDZ3 | Scrib-PDZ4 |
|---|---|---|---|---|---|
| TMIGD1 | DPHSETAL$_{COOH}$ | 18.17 ± 1.80 μM | NB | 9.12 ± 0.67 μM | NB |
| APC | GSYLVTSV$_{COOH}$ | 5.97 ± 1.80 μM | 35,94 ± 1.10 μM | 18.28 ± 3.33 μM | NB |
| β-PIX | PAWDETNL$_{COOH}$ | 3.3 ± 0.3 μM | 67,8 ± 7.9 μM | 14.5 ± 2.1 μM | NB |
| MCC | PHTNETSL$_{COOH}$ | 7.71 ± 0.50 μM | NB | 4.97 ± 0.90 μM | NB |
| Tax1 | KHFRETEV$_{COOH}$ | 7.8 ± 0.8 μM | 15.4 ± 1.2 μM | 9.1 ± 0.7 μM | 40.2 ± 3.1 μM |
| Vangl2 | RLQSETSV$_{COOH}$ | 24.49 ± 3.90 μM | 45,13 ± 4.92 μM | 40,16 ± 3.92 μM | NB |

*APC* adenomatous polyposis coli, *β-PIX* PAK-interacting exchange factor beta, *MCC* mutated in colorectal cancer, *NB* no binding, *PBM* PDZ domain binding motif, *Tax1* trans-activating transcriptional regulatory protein, *Vangl2* Vang-like protein 2. The PBM in the peptide sequences are underlined.

We next expressed the Scrib/PDZ-CAAX construct in TMIGD1-transfected HEK293 cells. Since CAAX motifs can target proteins to endomembranes such as the endoplasmic reticulum (ER) and the Golgi network[45], we reasoned that this construct could possibly recruit TMIGD1 to endomembranes. TMIGD1/f.l. but not TMIGD1/Δ5 co-localized with Scrib/PDZ-CAAX at endomembranes of transfected cells (Fig. 5b). Co-stainings of Scrib/PDZ-CAAX-transfected cells with markers for the ER (KDEL) and for the trans-Golgi network (TGN46) indicated that Scrib/PDZ-CAAX localized in the Golgi (Supplementary Fig. 4), suggesting that TMIGD1 was retained at the trans-Golgi network by the Scrib/PDZ-CAAX construct. These observations, thus, provided further evidence for an interaction of the two proteins in cells.

To address the question if TMIGD1 can recruit Scrib in polarized epithelial cells, we analyzed the localization of Scrib constructs in MDCKII cells, which are polarized epithelial cells derived from kidney distal tubules[46]. We have previously observed that ectopically expressed TMIGD1/f.l. is localized exclusively in the cytoplasm in MDCKII cells whereas a construct lacking the membrane-distal Ig-like domain (ΔD1-TMIGD1) was localized at the basolateral membrane[25]. We therefore used ΔD1-TMIGD1 MDCKII cells to analyze Scrib recruitment by TMIGD1 in polarized epithelial cells. ΔD1-TMIGD1 co-localized with endogenous β-catenin and Scrib at the basolateral membrane domain of MDCKII cells (Supplementary Fig. 5a). Ectopically expressed Scrib/LRR was localized exclusively in the cytoplasm of ΔD1-TMIGD1 cells, whereas Scrib/PDZ was present at the basolateral membrane domain (Supplementary Fig. 5b). These findings strongly suggest that TMIGD1 can recruit Scrib to the basolateral membrane domain in polarized epithelial cells as well.

Given the relevance of the Pro305 for Scrib membrane recruitment and its tumor-suppressive activity in cells[18,47], we tested the ability of TMIGD1 to recruit Pro305-mutated Scrib to cell-cell junctions. As opposed to Scrib/f.l. which efficiently localized at cell-cell junctions of HEK293 cells (Fig. 5a), Scrib/P305L did not localize to cell-cell contacts in HEK293 WT cells, which is consistent with previous observations in MCF10A cells[18,47]. However, in HEK293 cells expressing TMIGD1 but not in HEK293 cells expressing TMIGD1/Δ5, Scrib/P305L was efficiently recruited to cell-cell contacts (Fig. 6) indicating that TMIGD1 can restore cell-cell contact localization of Scrib/P305L by interacting with the PDZ domain module. Together, our findings identified TMIGD1 as a cell adhesion receptor that recruits Scrib to cell-cell contacts in epithelial cells. Our observations thus have implications for the understanding of the tumor-suppressing properties of these two proteins.

**Dominant-negative TMIGD1 impairs apical-basal polarity in MDCKII cells.** Since Scrib is a conserved cell polarity protein[15], its interaction with TMIGD1 suggested a possible role of TMIGD1 in the regulation of apical-basal cell polarity. To test this, we used MDCK cells which when grown in a three-dimensional matrix of collagen I or matrigel, develop into spheroids consisting of a lumen surrounded by a single layer of epithelial cells. Individual cells are polarized along the apical-basal polarity axis with an apical domain facing the lumen, a lateral domain in contact with adjacent cells, and a basal domain in contact with the ECM[48]. A failure in developing apical-basal polarity, for example after inhibition of cell polarity proteins CRB3, PATJ, PAR3 or aPKC, results in the development of multicellular aggregates instead of single lumen-containing cysts[49–53]. To test a role of TMIGD1 in cystogenesis we ectopically expressed the ΔD1-TMIGD1 mutant. Previous studies had shown that an analogous mutant of the TMIGD1-related Ig-SF family member JAM-A acts in a dominant-negative manner in the development of apical-basal polarity and cyst formation[54,55]. Control MDCKII cells developed normal cysts containing a single lumen surrounded by single-layered epithelium (Fig. 7a). Induced expression of ΔD1-TMIGD1 resulted in the formation of multicellular aggregates without lumen (Fig. 7a), a phenotype which strongly resembles the phenotype observed after depletion of Scrib in mammary gland-derived acinar epithelial cells[56] and in lung organotypic cultures[57]. Multilumenal cysts, which can also be observed after interfering with the function of some polarity proteins[58–60], were not observed in ΔD1/TMIGD1-expressing cells (Fig. 7a). To test if the interaction with Scrib could be responsible for the observed cyst phenotype, we performed cyst assays using MDCKII cells expressing a ΔD1-TMIGD1 construct that lacked the PBM (ΔD1-TMIGD1/Δ5). Expression of ΔD1-TMIGD1/Δ5 did not result in multicellular aggregates but in lumen-containing spheroids. However, these spheroids contained multiple lumens instead of the single lumen observed in control cells (Fig. 7b). Together, these observations suggest a role for TMIGD1 in the development of apical-basal polarity in polarized epithelial cells. They further support the notion that the interaction with Scrib could be responsible for this function of TMIGD1.

## Discussion

In this study we identify the cell adhesion molecule TMIGD1 as membrane anchor for the Scrib tumor suppressor protein. TMIGD1 and Scrib interact directly through a PDZ domain-mediated interaction that involves three out of the four Scrib PDZ domains. Importantly, after inactivating the endogenous mechanism of Scrib membrane recruitment — by deleting the Scrib LRR domain or by mutating Pro305 present in LRR12 – Scrib recruitment to the membrane depends on TMIGD1. Our findings thus identify a mechanism of Scrib membrane recruitment that depends on its PDZ domain region and that operates in addition to the LRR domain-dependent mechanism of membrane recruitment. Our findings also identify TMIGD1 as an adhesion

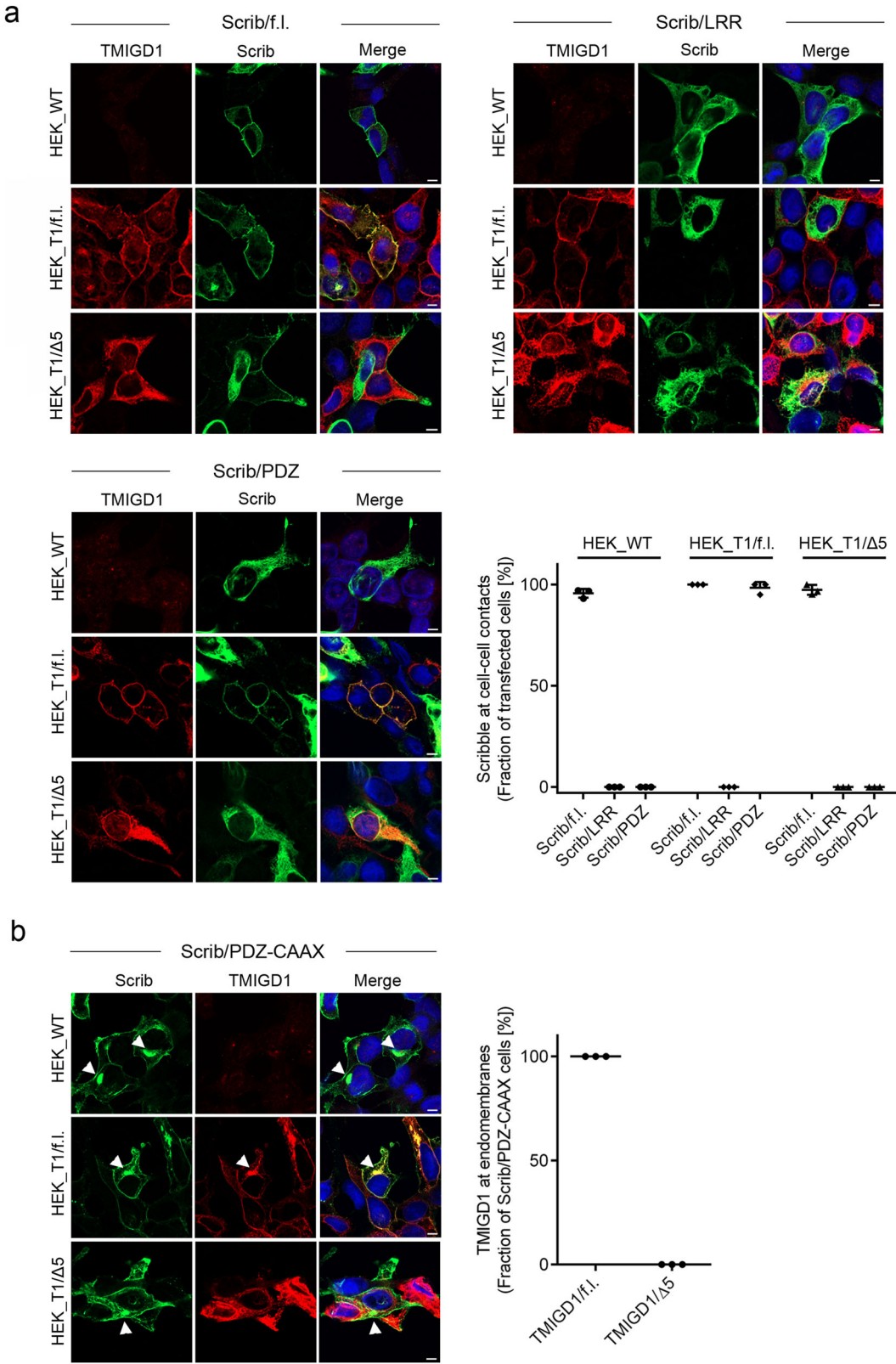

receiver localized at cell-cell contacts of polarized epithelial cells with the ability to recruit Scrib to the lateral membrane domain. Given that the tumor-suppressive activity of Scrib strongly depends on its membrane localization[16], these findings have important implications for tumor development.

Despite a strong interaction between recombinant TMIGD1 and Scrib constructs (Figs. 1b, 4b), the interaction between the full-length proteins expressed in cells was relatively weak (Figs. 1, 2). We explain this observation by the fact that Scrib can interact with a large number of interaction partners (summarized in refs. [15–17]). At least 37 proteins directly interact with the PDZ domains of Scrib, and in the majority of cases, i.e. 25, these interactions involve more than one PDZ domain[17]. It is very likely that some of these binding partners interact with one or

**Fig. 5 TMIGD1 recruits Scrib to cell-cell contacts. a** HEK293 cells, either untransfected (HEK_WT) or stably transfected with TMIGD1/full length (HEK_T1/f.l.) or with TMIGD1 lacking the PDZ domain motif (HEK_T1/Δ5) were transiently transfected with the indicated Scrib mutant constructs and stained for TMIGD1 and Scrib. Note that Scrib/f.l. localizes to cell-cell contacts in the absence of TMIGD1 (HEK_WT), but that Scrib/PDZ localization at cell-cell contacts depends on TMIGD1 and is mediated by the PDZ domain motif of TMIGD1. Quantification of Scrib recruitment to cell-cell contacts. The dot plot graph shows the fraction of cells with Scrib localization at cell-cell contacts. Scrib/f.l.: $n = 94$ (HEK_WT), $n = 81$ (HEK_T1/f.l.), $n = 76$ (HEK_T1/Δ5); Scrib/LRR: $n = 99$ (HEK_WT), $n = 64$ (HEK_T1/f.l.), $n = 63$ (HEK_T1/Δ5); Scrib/PDZ: $n = 91$ (HEK_WT), $n = 68$ (HEK_T1/f.l.), $n = 72$ (HEK_T1/Δ5); $N = 3$ independent experiments, represented by individual dots. Scale bars: 5 μm. **b** HEK293 cell lines described in (**a**) were transiently transfected with a Scrib/PDZ-CAAX construct and stained for TMIGD1 and Scrib. Note that the Scrib/PDZ-CAAX localizes to endomembrane structures (arrowheads), and that TMIGD1/f.l., but not TMIGD1/Δ5 co-localizes with Scrib/PDZ-CAAX at endomembranes. Right: Quantification of TMIGD1 co-localization with Scrib/PDZ-CAAX at endomembranes. The graph shows the fraction of Scrib/PDZ-CAAX-transfected cells with TMIGD1 localization at endomembranes. HEK_TMIGD1/f.l.: $n = 81$, HEK_T1/Δ5: $n = 76$. $N = 3$ independent experiments, represented by individual dots. Scale bars: 5 μm.

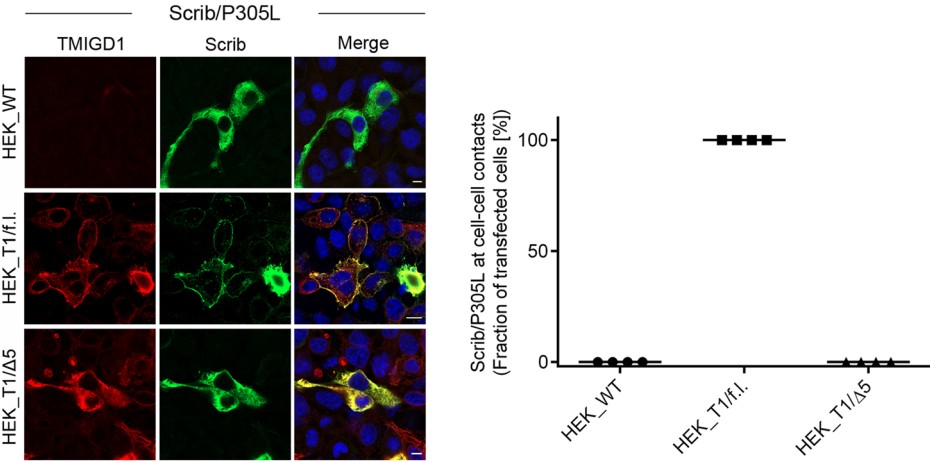

**Fig. 6 TMIGD1 recruits Scrib/P305L to cell-cell contacts.** HEK293 cells stably transfected with TMIGD1/full length (HEK_T1/f.l.) or with TMIGD1 lacking the PBM (HEK_T1/Δ5) were transiently transfected with a Scrib construct containing a Pro[305] to Leu[305] mutation (Scrib_P305L) and stained for TMIGD1 and Scrib. Quantification of Scrib recruitment to cell-cell contacts. The dot plot graph shows the fraction of cells with Scrib localization at cell-cell contacts. HEK_WT: $n = 138$; HEK_T1/f.l.: $n = 119$; HEK_T1/Δ5: $n = 111$; $N = 4$ independent experiments, represented by individual dots. Scale bars: 5 μm.

more Scrib PDZ domains with higher affinities than TMIGD1, thus outcompeting TMIGD1 from Scrib binding. For example, the affinity of the adenomatous polyposis coli (APC) PBM peptide for Scrib PDZ1 is approximately three-fold higher than that of the TMIGD1 PBM peptide ($K_D = 6.0$ μM vs $K_D = 18.17$ μM for APC vs TMIGD1, respectively[38]), suggesting that APC binding to Scrib PDZ1 will be favored over TMIGD1 binding. Another possibility to explain the weak interaction of TMIGD1 and Scrib in cells is that ligands which interact with Scrib could favor a conformation of Scrib that is unfavorable to TMIGD1 binding.

We have also observed that the recombinant PDZ2 domain of Scrib had no affinity towards the TMIGD1 C-terminal peptide (Fig. 4c, Table 2). On the other hand, inactivation of PDZ2 in a Scrib/PDZ1-4 construct resulted in a strong reduction in TMIGD1 binding in GST-pulldown assays (Fig. 4b), which suggested that PDZ2 contributes to TMIGD1 binding. We have presently no clear-cut explanation for this observation. One possibility would be that regions adjacent to PDZ2 influence PDZ2 ligand accessibility. As one example for such a scenario, Scrib PDZ3 and PDZ4 form a supramodule in which ligand binding to PDZ3 is enhanced by the linker region connecting PDZ3 and PDZ4[44]. As another example, in PSD-95, a scaffolding protein with three PDZ domains, a single SH3 domain and a guanylate kinase-like (GK) domain, PDZ3, SH3 and GK form a supertertiary structure in which ligand binding to PDZ3 influences subsequent homotypic and heterotypic complex formation[61,62]. It is not clear if similar mechanisms may apply to PDZ2 of Scrib. However, since GST-fusion proteins dimerize through the GST-domain[63], an allosteric mechanism involving a

supertertiary structure of the Scrib PDZ domain module[62] could provide one possible explanation for our observations.

Despite its physiological and, in particular, pathophysiological relevance, the mechanisms of Scrib membrane targeting are not fully understood. Anchoring at the lateral membrane domain of epithelial cells depends on the LRR-region[64–66]. Mutating Pro305 in the LRR12 to Leu (Scrib/P305L) abolishes membrane localization[47,56,66,67]. Thus, the LRR region of Scrib is a dominant driver of Scrib membrane localization. Importantly, ectopic expression of Scrib/P305L in cell lines disrupts cell polarity, suppresses c-myc-induced apoptosis, enhances Akt signaling, and fails to suppress Ras-MAPK-induced EMT[18,47,56,68]. Also, transgenic mice expressing Scrib/P305L in the mammary gland display hyperplastic growth in the mammary gland and develop mammary tumors[18]. These findings suggested that the function of Scrib as tumor suppressor depends on membrane localization, and that the LRR domain of Scrib is the dominant force for membrane localization.

Previous studies in Drosophila and in vertebrate MDCK cells, however, showed that Scrib constructs lacking the entire LRR region can also localize to cell-cell junctions[65,66,69], raising the question why the Scrib/P305L is not recruited by a PDZ domain-dependent mechanism. It is possible that replacing Pro305 with a hydrophobic Leu residue alters the conformation of the horseshoe-like structure of the LRR domain[70], which could abolish LRR-mediated interactions required for membrane recruitment[71], enhance interactions with a cytoplasmic protein, as demonstrated for ZDHHC7[67], or mask the PDZ domains of Scrib making these inaccessible for the many PDZ domain-dependent interaction partners[17]. Another possibility would be

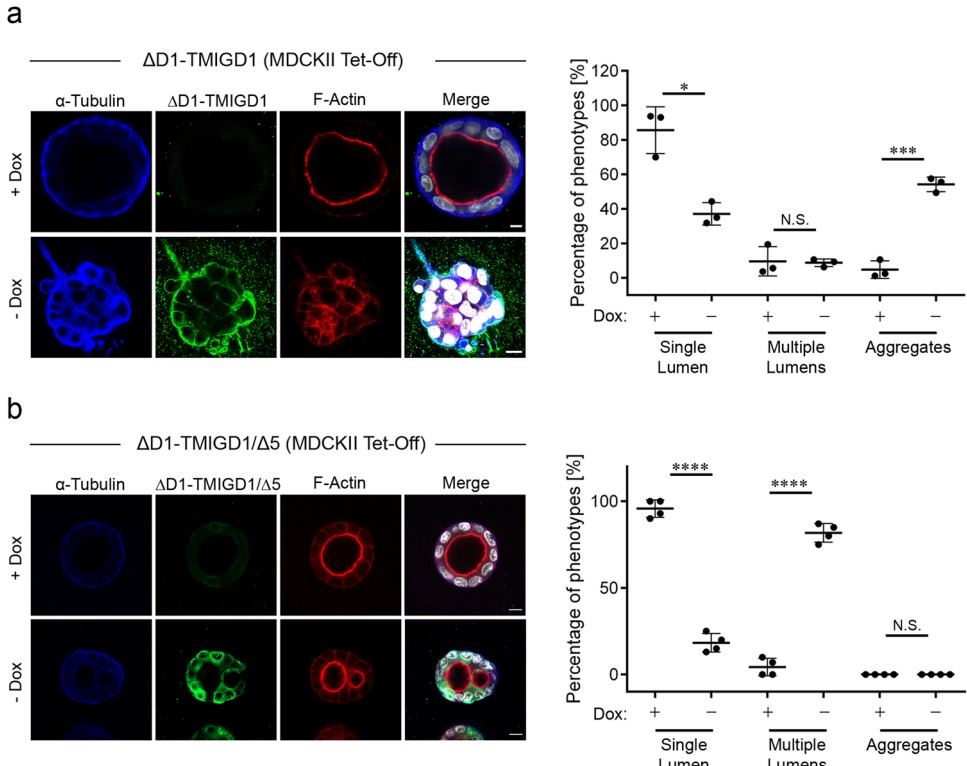

**Fig. 7 ΔD1-TMIGD1 impairs lumen formation in MDCKII cysts.** MDCKII cells stably transfected with a TMIGD1 construct lacking the membrane-distal Ig-like domain (ΔD1-TMIGD1) (**a**) or a TMIGD1 mutant lacking the PBM in addition to the membrane-distal Ig-like domain (ΔD1-TMIGD1/Δ5) (**b**) under a doxycycline (Dox)-regulated promoter were left uninduced (+ Dox) or were induced by Dox removal (- Dox), and cultured for 6 days in a three-dimensional collagen matrix. Cysts were stained for α-tubulin, TMIGD1 (anti Flag tag) and F-actin, DNA was stained with DAPI. Scale bars: 5 µm. The dot plots show the quantification of lumen formation. Cyst phenotypes were categorized into cysts with single lumen (Single Lumen), cysts with multiple lumens (Multiple Lumens), and multicellular aggregates without lumen (Aggregates). Statistical evaluation was performed with unpaired Student's *t* test. Data are represented as means ± SD (ΔD1-TMIGD1: *n* = 480, each + Dox and - Dox; *N* = 3 independent experiments; ΔD1-TMIGD1/Δ5: *n* = 63 (+ Dox), *n* = 62 (– Dox); *N* = 4 independent experiments). N.S. not significant, *$p < 0.05$, ***$p < 0.001$, ****$p < 0.0001$.

that the P305 mutation abolishes homodimerization, which is mediated through the LRR region[71], and that homodimerization is required for membrane targeting.

A role of TMIGD1 in recruiting Scrib to the lateral membrane is consistent with a predicted function of TMIGD1 as tumor suppressor protein. Several unbiased studies describe a strong and highly significant reduction of TMIGD1 gene expression in colon cancer in humans[72–75] with a progressive downregulation during disease progression[21,76]. Also, several studies found a significant downregulation of TMIGD1 gene expression in diseases associated with chronic intestinal inflammation such as Crohn's disease or inflammatory bowel disease[26,73,77], conditions that promote carcinogenesis[78]. In further support of a role of TMIGD1 as tumor suppressor, inactivation of the *Tmigd1* gene in mice results in a grossly altered morphology of intestinal tissues associated with intestinal adenoma formation and enhanced proliferative activity in intestinal crypts[27]. Finally, TMIGD1 expression has also been found to be downregulated in kidney cancer[24]. Together, these findings support the hypothesis that the function of TMIGD1 as tumor suppressor could be linked to its ability to recruit Scrib to the membrane. It will thus be important to study a cause-and-effect relationship between loss of TMIGD1 expression and Scrib mislocalization during cancer development.

## Methods

**Cell culture and transfections.** HEK293T cells (ATCC-CRL-2316) were grown in DMEM containing 10% FCS, 1% NEAA, 2 mM L-Glu, 100 U/ml Pen/Strep. Stably transfected HEK293 cells expressing TMIGD1/f.l. or TMIGD1/Δ5 were generated by electroporation (0.25 kV, 950 µF) and cultured in growth media supplemented with G418 (800 µg/ml). Positive clones were identified by Western blot analysis and immunofluorescence microscopy. Transient transfections of cDNAs were performed using Lipofectamine 2000 (ThermoFisher Scientific, #11668-019) and X-Fect (Xfect™ Transfection Reagent, TakaraBio-Clontech #631318), according to manufacturer's instructions. MDCK II Tet-Off cell lines (TakaraBio-Clontech, St-Germain-en-Laye, # 630913/631138) were maintained in DMEM containing 10% FCS, 2 mM L-glutamine, 100 U/ml Pen/Strep, 100 µg/ml G418 and 1 µg/ml puromycin. MDCK II Tet-Off cells stably expressing a human TMIGD1 mutant lacking the membrane-distal Ig-domain (ΔD1-TMIGD1) were generated by electroporation and subsequent selection by growth in DMEM medium supplemented with 150 µg/ml hygromycin and 50 ng/ml doxycycline (Dox). Expression of transgenes was induced by transferring cells into medium lacking Dox using tetracycline-free FCS (BD Biosciences)[25]. All cell lines were routinely tested and found to be negative for mycoplasma contamination.

**Expression vectors.** The following constructs were used. TMIGD1 constructs in pcDNA3 (Invitrogen): hTMIGD1 full length (AA 1-262), hTMIGD1 lacking the PDZ domain binding motif (hTMIGD1/Δ5, AA 1-258). TMIGD1 constructs in pKE576hyg: Flag-tagged human TMIGD1 lacking the membrane-distal Ig-like domain (hΔD1/TMIGD1, AA 115 - 262)[25]. GST-tagged TMIGD1 constructs in pGEX-4T-1 (GE Healthcare): GST-TMIGD1 (hTMIGD1) cytoplasmic tail (AA 242-262); GST-TMIGD1/Δ5: hTMIGD1 cytoplasmic tail lacking the PDZ domain binding motif (AA 242-257). TMIGD1 constructs in yeast-two hybrid vector pBTM116[79]: pBTM116-TMIGD1 (cytoplasmic tail of hTMIGD1, AA 241-262). TMIGD1-Fc fusion constructs in pcDNA3-hIgG: hTMIGD1-Fc (extracellular domain of hTMIGD1, AA 1 - 222).

Human Scribble (Uniprot accession number Q14160) constructs in pcDNA3:His/Xpress[41]: hScrib/PDZ-WT (AA 616-1490), hScrib/PDZ-mut1 (AA 616-1490_L$_{738}$AG$_{739}$E), hScrib/PDZ-mut2 (AA 616-1490_L$_{872}$AG$_{873}$E), hScrib/PDZ-mut3 (AA 616-1490_L$_{1014}$AG$_{1015}$E), hScrib/PDZ-mut4 (AA 616-1490_L$_{1111}$AG$_{1112}$E). Human Scrib constructs in pKE081myc (N-terminal myc tag)[80]: hScrib/WT (AA 2-1630), hScrib/P305L (AA 2-1630, Pro305Leu), hScrib/LRR (AA 2-727), hScrib/PDZ (AA 665-1630), hScrib/PDZ-CAAX (AA 665-1630 with C-terminal CAAX box (GCMSCKCVLS) derived from H-Ras). Human Scrib

PDZ domain constructs in pGEX-6P-3 (GE Healthcare) or pGIL-MBP[81]: hScrib/PDZ1 (AA 728-815), hScrib/PDZ2 (AA 833-965), hScrib/PDZ3 (AA 1005-1094), hScrib/PDZ4 (AA 1099-1203). Erbin constructs in pEGFP-GW (kindly provided by Dr. Jean-Paul Borg, Inserm, CNRS, Marseille, France): hErbin/WT (AA 1-1412). Lano constructs in pcDNA-HA (kindly provided by Dr. Jean-Paul Borg, Inserm, CNRS, Marseille, France): hLano/WT (AA 1 – 524).

**Antibodies and reagents.** The following antibodies were used in this study: rabbit pAb anti-TMIGD1 (SA #HPA021946, IF 1:500); mouse mAb anti-α-Tubulin (SA, clone B-5-1-2, #T5168, IF 1:500, WB 1:10.000); rabbit pAb anti-TGN46 (abcam #ab50595, IF 1:500); rabbit pAb anti-KDEL (ThermoFisher Scientific #PA1-013, IF 1:2.500); mouse mAb anti-β-catenin (BD-TransductionLabs #610153, IF 1:500); mouse mAb anti-Scribble (SantaCruz #sc-55543, IF 1:500); goat pAb anti-Myc (SantaCruz #sc-789G, IF 1:500, WB 1:500); mouse mAb anti-Myc 9E10 [82] (IF 1:500, WB 1:500); mouse mAb anti-GFP (Takara #632375, WB 1:500); rabbit pAb anti-Hemagglutinin (HA) (SA, #H6908, WB: 1:500); rabbit pAb anti-Flag (SA, #F7425, IF 1:500); mouse mAb anti-6xHis (proteintech #66005-1-Ig, IF 1:500). Rabbit anti-TMIGD1 pAb Affi1662/1663 (IF 1:500, WB 1:500) was generated by immunizing rabbits with a fusion protein consisting of the extracellular domain of hTMIGD1 fused to the Fc region of human IgG[28]. The antibodies were affinity-purified by adsorption at the antigen covalently coupled to cyanogen bromide (CNBr)-activated sepharose beads (Amersham Biosciences Europe, Freiburg, Germany). Antibodies directed against the Fc part were depleted by adsorption at human IgG coupled to CNBr-activated sepharose beads. Affinity-purified antibodies were dialyzed against PBS. Secondary antibodies and fluorophore-conjugated antibodies: Fluorophore-conjugated antibodies for Western blotting: IRDye 800CW Donkey anti-Rabbit IgG (LI-COR Biosciences #926-32213, WB 1:10.000), IRDye 680CW Donkey anti-mouse IgG (LI-COR Biosciences #926-68072, WB 1:10.000). Fluorophore-conjugated secondary antibodies for ICC: Donkey anti-Mouse IgG (H + L) Alexa Fluor 594 (ThermoFisher Scientific #A-21203); Donkey anti-Rabbit IgG (H + L) Alexa Fluor 594 (ThermoFisher Scientific #A-21207); Donkey anti-Rabbit IgG(H + L) Alexa Fluor 488 (ThermoFisher Scientific #A-21206); Donkey anti-Mouse IgG (H + L) Alexa Fluor 488 (Dianova/Jackson ImmunoResearch Europe Ltd #715-545-150); Donkey anti-Mouse IgG (H + L) Alexa Fluor 647 (Dianova/Jackson ImmunoResearch Europe Ltd #715-605-150) -conjugated, highly cross-adsorbed secondary antibodies (all Alexa Fluor-conjugated secondary antibodies for IF were used in a dilution of 1:800). The following peptides were used for crystallization and ITC: -DPHSETAL, -DPHSEAAA; peptides were obtained from Genscript. The following reagents were used: Dox (SA #D9891), collagen type I (rat tail type 1 collagen, Advanced Bio-Matrix #5163), 2,4,diamidino-2-phenylindole (DAPI, SA # D9542).

**Yeast two-hybrid screen.** Yeast two-hybrid screening experiments were performed essentially as described[80]. Briefly, the Saccharomyces cerevisiae reporter strain L40 expressing a fusion protein between LexA and the cytoplasmic tail of TMIGD1 (AA 241–262) was transformed with 250 μg of DNA derived from a day 9.5/10.5 mouse embryo cDNA library[83] according to the method of Schiestl and Gietz[84]. The transformants were grown for 16 h in liquid selective medium lacking tryptophan and leucine (SD-TL) to maintain selection for the bait and the library plasmid, then plated onto synthetic medium lacking tryptophan, histidine, uracil, leucine, and lysine (SD-THULL) in the presence of 1 mM 3-aminotriazole. After 3 days at 30 °C, large colonies were picked and grown for an additional three days on the same selective medium. Plasmid DNA was isolated from growing colonies using a commercial yeast plasmid isolation kit (DualsystemsBiotech, Schlieren, Switzerland). To segregate the bait plasmid from the library plasmid, yeast DNA was transformed into E. coli HB101, and the transformants were grown on M9 minimal medium lacking leucine. Plasmid DNA was then isolated from E. coli HB101 followed by sequencing to determine the nucleotide sequence of the inserts.

**Immunoprecipitation and western blot analysis.** For immunoprecipitations (IPs), cells were lysed in lysis buffer (25 mM TrisHCl, pH 7.4, 1% (v/v) Nonidet P-40 (NP-40, AppliChem, Darmstadt, Germany), 150 mM NaCl, protease inhibitors (Complete Protease Inhibitor Cocktail; Roche, Indianapolis, IN), 5% glycerol (AppliChem #A2926) and phosphatase inhibitors (PhosSTOP™, Roche, Indianapolis, IN), 2 mM sodium orthovanadate) for 20 min with overhead rotation at 4 °C followed by centrifugation (15.000 rpm, 20 min at 4 °C). Postnuclear supernatants were incubated with 3 μg of antibodies coupled to protein A– or protein G–Sepharose beads (GE Healthcare, Solingen, Germany) for 4 h at 4 °C. Beads were washed five times with lysis buffer, bound proteins were eluted by boiling in 3x SDS-sample buffer/150 mM DTT. Eluted proteins were separated by SDS–PAGE and analyzed by Western blotting with near-infrared fluorescence detection (Odyssey Infrared Imaging System Application Software Version 3.0 and IRDye 800CW-conjugated antibodies; LI-COR Biosciences, Bad Homburg, Germany). All IP and Western blot experiments shown in this study are representative for at least three independent experiments.

**Protein expression and purification of recombinant Scrib PDZ constructs.** All proteins were overexpressed in E. coli BL21 (DE3) CodonPlus or E. coli BL21 (DE3) pLysS cells (BIOLINE) via manual induction using 0.5 mM isopropyl 1-thio-β-d-galactopyranoside either as Glutathione S-transferase or Maltose Binding Protein fusions. Recombinant protein purification and tagged protein cleavage were carried out as described[35]. Purified Scribble PDZ domains were subsequently dialysed into 25 mM Tris-HCl pH 8.0, 150 mM NaCl, 5 mM TCEP (Tris(2-carboxyethyl)) phosphine hydrochloride (Buffer A) for downstream applications.

**GST pulldown experiments.** In vitro binding experiments were performed with recombinant GST-TMIGD1 fusion proteins purified from E.coli and immobilized on glutathione-Sepharose 4B beads (Life Technologies #17-0756-01). Purification of GST fusion proteins was performed as described[80]. For the analysis of direct protein interactions, Scrib constructs were translated in vitro using the TNT T7-coupled reticulocyte lysate system (Promega Corp., Madison, WI) in the presence of [35][S]-labeled methionine (Hartmann Analytic GmbH, Braunschweig, Germany) as described by the manufacturer. 10 μl of the translation reactions were incubated with 3 μg of immobilized GST fusion protein for 2 h at 4 °C under constant agitation in buffer B (10 mM Hepes-NaOH (pH7.4), 100 mM KCl, 1 mM MgCl₂, 0.1% Triton X-100). After 5 washing steps in buffer B bound proteins were eluted by boiling for 5 min in SDS sample buffer, subjected to SDS-PAGE and analyzed by fluorography. All in vitro binding experiments shown in this study are representative for at least three independent experiments.

**MDCK II cyst assays.** MDCK cyst assays were performed essentially as described[54,85]. Briefly, MDCK II cells were seeded as single cell suspension ($3.4 \times 10^4$ cells/ml) in 0.18% type I collagen from rat tail (BD Biosciences). After 3–6 d, the gels were washed in PBS, subjected to collagenase treatment (100 U/ml collagenase VII (Sigma), 15 min, RT) and fixed (4% paraformaldehyde/PBS, 30 min, RT). Cells were permeabilized by treatment with 0.25% Triton X-100/PBS (30 min, RT) and washed extensively with 2% goat serum/PBS (1 h, RT). Incubations with primary and fluorochrome-conjugated secondary antibodies were performed in 2% goat serum/PBS for a minimum of 12 h at 4 °C. After extensive washing, the gels were mounted on glass coverslips using Kaiser's glycerol gelatine (Merck, Darmstadt, Germany). Cysts were analyzed using a confocal microscope as specified in the Immunocytochemistry and Immunofluorescence microscopy paragraph.

**Immunocytochemistry and immunofluorescence microscopy.** For immunofluorescence microscopy, cells were grown on collagen-coated glass slides. Cells were washed with PBS and fixed with 4% paraformaldehyde (PFA, Sigma-Aldrich) for 7 min or with -20 °C-cold MeOH for 5 min. To detect intracellular proteins, PFA-fixed cells were incubated with PBS containing 0.2% Triton X-100 for 15 min. Cells were washed with 100 mM glycine in PBS, blocked for 1 h in blocking buffer (PBS, 10% FCS, 0.2% Triton X-100, 0.05% Tween-20, 0.02% BSA) and then incubated with primary antibodies in blocking buffer for 1 h at room temperature (RT) or overnight at 4 °C. After incubation, cells were washed three times with PBS and incubated with fluorochrome (AlexaFluor488, AlexaFluor594 and Alexa-Fluor647)-conjugated, highly cross-adsorbed secondary antibodies (Invitrogen) for 2 h at RT protected from light. F-Actin was stained using phalloidin-conjugates (TRITC) and cytoPainter iFluor-647, DNA was stained with 4,6-diamidino-2-phenylindole (DAPI, SA). Samples were washed three times with PBS and mounted in fluorescence mounting medium (Mowiol 4-88, SA). Immunofluorescence microscopy was performed using the confocal microscope LSM 800 Airyscan (Carl Zeiss, Jena, Germany) equipped with the objective Plan-Apochromat x 63/1.4 oil differential interference contrast (Carl Zeiss). Image processing and quantification was performed using ImageJ and Zen 2 (Blue Edition, Carl Zeiss) softwares. For quantification of Scrib recruitment to the lateral membrane domain by TMIGD1, at least 60 and maximally 90 cells per condition (derived from 3 independent experiments) were analyzed. Statistical analysis was performed using unpaired Student`s t test, data is plotted as means ± standard deviation (SD). P values: *$P < 0.05$, **$P < 0.01$, ***$P < 0.001$, ****$P < 0.0001$.

**Isothermal titration calorimetry.** All titration experiments were performed with the purified PDZ domains in Buffer A against C-terminally derived 8-mer human TMIGD1 peptide (DPHSETAL; UniProt ID: Q6UXZ0) and a mutant TMIGD1 peptide with the sequence DPHSEAAA. Protein concentrations were measured using a NanoDrop 2000 spectrophotometer (Thermo Fisher scientific)[39,86]. Each domain was used at a concentration of 75 μM and the peptide concentration was 900-1100 μM. Titrations were performed at 25 °C with a stirring speed of 750 rpm using the Microcal™ NanoITC200 system (GE Healthcare) with a total of 20 injections. Raw data was processed with MicroCal Origin version 7.0 software (OriginLab™ Corporation) using a one-site binding model. The affinity data of each protein were compared to the positive control of a synthetic pan-PDZ binding peptide known as super peptide (RSWFETWV)[87].

**Protein crystallization.** Scrib/PDZ1:TMIGD1 PBM peptide complexes were prepared for crystallisation trials as described[88]. The protein:peptide complexes were resuspended in buffer A at a molar ratio of 1:4. Using an in-house Gryphon LCP liquid dispenser (Art Robbins Instruments), initial sparse matrix crystallisation trials were performed in 96-well sitting drop trays (Swissci AG, Neuheim, Switzerland) with 0.2 μl of protein sample and 0.2 μl reservoir per drop. Subsequent

crystal optimisations were carried out in 24-well Limbro plates (Hampton Research). All crystallisation trials were performed at 20 °C. Human Scribble PDZ1:Human TMIGD1 PBM crystals were obtained at 8 mg/ml in 24% w/v PEG 1500, flashed cool in 20% ethylene glycol.

**Diffraction data collection and structure determination**. All diffraction data were collected on the MX2 beamline at the Australian Synchrotron using an Eiger detector (Dectris, Baden-Dättwil, Switzerland) with an oscillation range of 0.1° per frame using a wavelength of 0.9537 Å. Diffraction data was integrated using DIALS and scaled with AIMLESS[89–91]. The structure of Scribble/PDZ1:TMIGD1 peptide complex was solved by molecular replacement with Phaser using human Scribble PDZ1:human APC (PDB ID: 6XA8)[40] as the search model[92]. The molecular replacement solution was manually rebuilt using Coot and refined with PHENIX[93,94]. Data collection and refinement statistics are summarised in Supplementary Table 1. Final images of Scrib/PDZ1:TMIGD1 peptide complex were generated using the PyMOL molecular graphic system, version 1.8.6 (Schrödinger, LLC, New York, USA) and all software was accessed through SBGrid suite[95]. All raw diffraction images were deposited on the SBGrid Data Bank[96] using the PDB accession code 8CD3.

**Statistics and reproducibility**. Results are expressed either as arithmetic means ± SD as indicated. To test the normality of data sample distributions, the D'Agostino-Pearson normality test was used. Data were statistically compared using unpaired, two-tailed Student's $t$ test, or probed for being statistically different from a fixed value using One sample $t$ test. Statistical analyses were performed using GraphPad Prism version 6 (GraphPad Prism 6.0 Software, San Diego, CA). P-values are indicated as follows: $*P < 0.05$, $**P < 0.01$, $***P < 0.001$ and $****P < 0.0001$. For each statistical analysis data derived from at least three independent experiments were used. Experiments were considered independent when they were derived from distinct samples and thus reflected biological replicates[97]. Sample sizes are indicated for all experiments in the respective figure legends.

**Reporting summary**. Further information on research design is available in the Nature Portfolio Reporting Summary linked to this article.

## Data availability

Data supporting the findings of this manuscript are available from the corresponding authors upon reasonable request. Numerical source data for all graphs is available in the Supplementary Data 1 file. Uncropped images of gels and blots are shown in Supplementary Fig. 6. The coordinate files for the Scrib/PDZ1:TMIGD1 peptide complex were deposited at the Protein Data Bank (PDB) using accession code 8CD3.

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

## Acknowledgements

We gratefully acknowledge the help of Dr. Jean-Paul Borg (Center der Recherche en Cancérolgie de Marseille, Aix Marseille University, Inserm, CNRS, Marseille, France) who provided us with expression vectors encoding Erbin and Lano/LRRC1. We also thank Frauke Brinkmann (Institute of Medical Biochemistry, ZMBE, University

Muenster) for excellent technical assistance. We would like to thank Miranda Thomas (International Center for Genetic Engineering and Biotechnology, Trieste, Italy) for helpful comments on the manuscript. We thank the staff at the MX beamlines at the Australian Synchrotron for help with X-ray data collection, and the Comprehensive Proteomics Platform at La Trobe University for core instrument support. This research was undertaken in part using the MX2 beamline at the Australian Synchrotron, part of ANSTO, and made use of the Australian Cancer Research Foundation (ACRF) detector. This work was supported by grants from the Deutsche Forschungsgemeinschaft (EB 160/8-1; EXC 1003-CiMIC) to KE, the National Health and Medical Research Council Australia (Project Grant APP1103871 to MK, POH; Senior Research Fellowship APP1079133 to POH) and La Trobe University (Research focus area "Understanding Disease" project grant and scholarship to JCM and AJ).

## Author contributions
E.-M.T., C.H., M.K., P.O.H. and K.E. designed and conceived the study. E.-M.T., C.H., J.C.M., A.J. and B.E.M. performed experiments. L.B. provided reagents. E.-M.T., C.H., V.G., M.K., P.O.H. and K.E. analyzed the data. E.-M. T., M.K., P.O.H. and K.E. wrote the manuscript.

## Funding

## Competing interests
The authors declare no competing interests.
