## [Peer Review File · Communications Biology]

Reviewers' comments:

Reviewer #1 (Remarks to the Author):

In the manuscript submitted by Thüring et al., the authors identified a previously unknown interaction between Scrib and a plasma membrane protein called TMIGD1. They highlight a mechanism of Scrib recruitment at the plasma membrane through its PDZ domain. The authors dissect biochemically the association between Scribble and TMIGD1 and demonstrate that this interaction is mediated by the Scrib PDZ domains, PDZ1 being the most prominent interacting domain out of the 4 PDZs of Scrib. To support this finding, the authors have also generated a crystal structure of the PDZ1 domain of Scrib with the C-terminus peptide of TMIGD1 encompassing the PDZ binding motif. Using transfection assays, they found that the recruitment of Scrib deleted of its LRR domains to the plasma membrane can be triggered by TMIGD1 full length but not by TMIGD1 devoided of its PDZ binding motif. As both Scrib and TMIGD1 have a tumor suppressing role in cancer, they explore their functional interaction in a 3D cellular model. Based on the perturbed 3D morphology following expression of TMIGD1 deleted of its Scrib binding site, they conclude that this phenotype is due to the loss of Scrib-TMIGD1 interaction.

Overall the paper characterized a novel interaction between Scrib and TMIGD1; however it fails to demonstrate the direct functional relationship between these two proteins. Intriguingly, the authors mention that many epithelial cell lines have no expression of TMIGD1 (Caco2, MDCK); however Scrib is perfectly basolateral in these cells suggesting that TMIGD1 is not mandatory for Scrib membrane localization.

As mentioned in the paper, previous works have shown that the LRR and the PDZ domains of Scrib are required for membrane localization. To show a prominent role of TMIGD1 in Scrib PDZ-mediated localization, the authors have to express the full length Scrib LRR mutant and rescue its membrane localization with TMIGD1.

Fig 1A. There is no mention of the other clones, including possibly other PDZ proteins than Scrib, that were probably selected by the two hybrid screen.

Fig 1C. The band corresponding to endogenous Scrib is not clear in the TMIGD1 immunoprecipitation.

Fig 1D. The rationale of using an inhibitor of UBE1 is not clear and the authors do not explain why this experiment has been performed. The authors should have worked with endogenous proteins to demonstrate the importance of UBE1 in the processing of Scrib and TMIGD1.

Fig 5. HEK293 cells are not really epithelial cells and are not a suitable model to study cell-cell adhesion. I suggest to the authors to perform these experiments using polarized epithelial cells such as MDCK cells. Moreover, there is no characterization of the basolateral colocalization of Scrib and TMIGD1.

Fig 6. The results are interesting, however as TMIGD1 has probably several PDZ interactors, it does not demonstrate the direct functional relationship between Scrib and TMIGD1. Expression of a TMIGD1 mutant with a specific mutation in its C-terminus (based on the structure and lying outside of the last four residues of the PDZ binding site) abrogating the Scrib interaction could add some value to the assay.

Reviewer #2 (Remarks to the Author):

This paper investigates the interaction between TMIGD1 and Scribble (Scrib), proteins involved in cell adhesion and polarity. The authors show, using yeast-two hybrid and GST-pulldown experiments, that TMIGD1 directly interacts with the first PDZ domain of Scrib, and this interaction is dependent on the C-terminal PBM of TMIGD1. Co-immunoprecipitation experiments indicate that Scrib is present in TMIGD1 full-length precipitates but not in TMIGD1/ Δ 5 precipitates, confirming that the interaction is

PBM-dependent in cells. The authors also demonstrate that both TMIGD1 and Scrib are subject to proteasomal degradation in HEK293 cells, and that the interaction is enhanced by the inhibitor of Ubiquitin-activating enzyme E1. Scrib mutant constructs show that the interaction is specifically dependent on the PDZ domain region of Scrib, that the 1st, 2nd, and 3rd domains have affinity, and that TMIGD1 does not interact with other LAP family members. The authors determine a crystal structure of the Scrib PDZ1 domain in complex with the C-terminal region of TMIGD1, showing the interaction is mediated through a canonical PDZ interaction. Finally, they observe Scrib localization in cells to conclude that TMIGD1 can recruit Scrib to the cell membrane in a PBM dependent manner, and that Scrib can correspondingly influence TMIGD1 localization. Overall, the paper provides some insight into a newly identified interaction between TMIGD1 and Scrib and considers the implications for cell polarity and adhesion. The crystal structure and in vitro biochemistry experiments are well performed and support the conclusions. Conversely, the imaging experiments open several questions that would need to be answered before publication. Specific concerns are:

1. The low affinity of the PDZ1-TMIGD1 complex suggests this PBM likely interacts with other PDZ domains, confirmed by its binding to 3/4 Scrib domains. Were there other PDZs found in the Y2H? This would help answer questions of specificity.
2. The interaction between recombinant TMIGD1 and endogenous Scrib is extremely weak, despite the PBM protein being expressed at high levels (also true with recombinant full length Scrib in Fig 2). As there are many known binding partners of the Scrib PDZ domains it is important to contrast the TMIGD1 interaction with others. Indeed, this is the weakest element of the manuscript – there was no attempt to inform us of all known PDZ1/PDZ3 interactions and contrast the TMIGD1 complex with alternatives. Do co-IPs of endogenous Scrib with other partners containing PBMs result in a more robust interaction?
3. The conclusions of the TAK-243 inhibitor experiment were that both proteins are modulated by ubiq/degradation. The levels in treated/untreated do not look significantly different to me, and to make this conclusion requires quantitation.
4. The inclusion of Scrib/PDZ-Caax was not well rationalized. What was the purpose of this experiment?
5. 2 alternative LAP proteins were assayed for TMIGD1 interaction, but the authors did not inform us of how many proteins are in the LAP family or how representative these two members are of the LAP family as a whole.
6. The crystal structure is high resolution and was well refined. To place it in context with previous work, it would be informative to compare the major PDZ1 binding determinants with other PDZ1-peptide complex structures.
7. How much GST proteins were used for the pull downs in Fig 4b? Seems like very small amounts to get such a robust binding when all previous interactions were so weak.
8. The rationale for observing loss of TMIGD1 binding with the PDZ2 mutant but not detecting an in vitro interaction (ITC) is not clear to me. The 4 PDZ domains of Scrib have been extensively studied, and there is no known intramolecular regulation of the PDZ2 domain. The paper cited as evidence of cooperation (alpha1D-adrenergic receptor) uses BLI and is not truly measuring cooperative binding, but rather avidity of multiple domains with overlapping specificity. How do the authors envision a cooperative mechanism impacting PDZ2 affinity for TMIGD1 in the pull down experiment but they observe no in vitro binding of the domain alone?
9. Was the Scrib PDZ3 picked up by Y2H? If not, was there a construct in the cDNA library that encompassed this region?
10. More effort should be made to place the findings in context with the extensive Scrib PDZ literature. How many PDZ1 ligands are known and what are their affinities? How does the TMIGD1 affinity compare?
11. The imaging experiments are the weakest component of the manuscript and should be improved. First, the LRR of Scrib has long been considered the membrane recruitment region of this protein (as discussed extensively by the authors in the discussion). Why does the Scrib LRR here not localize to

the membrane? For TMIGD1, the protein has a transmembrane domain but why does deletion of the final 5 residues of the cytoplasmic domain alter its localization? Should it not remain on the membrane? The authors state that "TMIGD1 as an adhesion receptor localized at cell-cell contacts of polarized epithelial cells which recruits Scrib to the lateral membrane domain" but there is no evidence of either in this manuscript. I see no evidence that TMIGD1 is at cell-cell contacts, only that it can be membrane localized. In polarized epithelial cells does it colocalize with markers of tight or adherens junctions? It appears more apical in the cysts presented here. And for evidence that Scrib is recruited to the lateral membrane we would need to see confocal data, z stacks, that this is true. The HEK images seem to show it is membrane localized. Finally, how is it that full length Scrib in these cells is at the membrane, but the LRR and PDZ regions alone are not. What is major recruitment factor for Scrib membrane localization in HEK cells?

12. The PDZ1/2/3 mutants should be included in the imaging analysis to verify mutational inactivation of these modules disrupts membrane recruitment.

13. The MDCK cyst experiments show that expression of TMIGD1 lacking the Ig domain induces aggregates and significant loss of well-formed cysts with a single lumen. This system, however, was not exploited to determine whether this has anything to do with Scrib or even with the PBM of TMIGD1. It is therefore unclear what this experiment brings to the current manuscript. Why is the $\Delta D1$ variant of TMIGD1 used here? What happens with full length TMIGD1? Also, an experiment with the ΔPBM variant would inform if the phenotype is dependent on binding to a PDZ domain. Finally, rather than use the red channel for actin there is an opportunity to stain for endogenous Scrib, which is highly expressed in these cells. Some further experiments here are required to support the conclusions.

14. Finally, relevance of TMIGD1 recruitment of Scrib is not completely clear to me. TMIGD1 is expressed only in gastrointestinal cells and even in these cells is difficult to detect at the protein level. The authors need to clarify how important they believe TMIGD1 is to normal Scrib function – or do they contend this interaction is relevant only in cancer cells? How do they place it in context of all the other Scrib PDZ binding partners? The designation of TMIGD1 as an 'adhesion receptor' localized at cell-cell contacts needs further validation as well, as I could not find strong data indicating this in the literature.

Reviewer #3 (Remarks to the Author):

Thuring et al., present a manuscript that follows up on a hit from a yeast two hybrid protein-protein interaction between the cell polarity protein Scribble and the cell adhesion molecule TMIGD1. They first validate the yeast two-hybrid result by showing that TMIGD1 interacts with Scribble in vitro before showing they weakly interact when expressed at high levels in cells (through preventing their degradation). They subsequently present data showing that the interaction is dependent on a PDZ binding motif (PBM) at the carboxy terminal of TMIGD1 with the PDZ portion specifically of Scribble, and not its relatives Erbin or Lano. Crystal structure analysis of the TMIGD1 PBM in complex with Scribble PDZ1 shows amino acid specific interaction in the PDZ binding pocket. Further in vitro binding assays show that TMIGD1 binds to PDZ1 and PDZ3 of Scribble. Lastly, Thuring and colleagues transiently express tagged versions of the TMIGD1 and Scribble in HEK293 and MDCK cells to understand in vivo dynamics of Scribble membrane recruitment and the role of TMIGD1 in cell polarity and luminogenesis.

Overall, the manuscript presents an intriguing finding, that the cell adhesion molecule TMIGD1 binds to Scribble PDZ domains 1 and 3. The authors assert this novel interaction between the two proteins forms the basis of Scribble membrane recruitment. They provide strong evidence from in vitro and gain of function in vivo protein interaction studies that couple well with atomic resolution data for the

interaction. However, the *in vivo* studies are lacking the necessary evidence to make the bold claim that this interaction is sufficient to explain how Scribble is recruited to the membrane in normal cells or that lack of this interaction is a driving cause of tumorigenesis. Below are the major concerns the authors should consider addressing before publication. The bulk of the concerns deal with over interpretation of published work and gathered data.

Comments for major concerns:

The title of the manuscript is misleading and over-interprets the data presented. Although the data show that Scribble can interact with TMIGD1, its relevance in a wide range of normal cells and tissues, or the lack of interaction in disease, is not deeply explored (see below). Data from gain of function experiments show that certain forms of truncated Scribble can be recruited to the membrane. A title change that better reflects the data presented is necessary.

The authors make the claim that the mechanism of Scribble membrane localization is not well understood (see lines 40-4, 92-93). This is not true. The full molecular mechanism of Scribble membrane localization may not be entirely understood especially as there may be mechanisms that vary between organisms, cell-types, and other aspects of biological context. The authors cite the Bonello and Peifer review with a whole section (Fig 2) which contains multiple examples of the known mechanisms for Scribble membrane interaction. The bold claim that membrane localization is unclear is not true. Binding to TMIGD1 may be another mechanism that is certainly context specific (since TMIGD1 is only expressed in a few cell types). The authors should reframe the overarching rationale. Finding the binding partners of Scribble is in itself interesting. They should also revisit lines 98-99, which makes the assumption that it is a cell adhesion molecule that Scribble requires. Why an adhesion molecule versus any other lipid molecule, transmembrane protein or cortex binding protein?

The assertion that it is the membrane localization of Scribble that prevents tumors, versus its cytoplasmic localization that promotes tumors, is not well reasoned. Statements on lines 39-40 and 89-91 reframe the original findings from Refs 17-19 that tumors resulting from cytoplasmic localization of Scribble mean that membrane localization is required. It is possibly true that the original finding of tumors that result from the P305L mutation has nothing to do with the membrane localization but rather sequestration of tumorigenic factors in the cytoplasm. That idea being that the cytoplasm localization could also be tumorigenic. The authors should reframe their argument to fit the data from the references they cite. In fact those papers also over-express membrane miss-localization versions of Scribble, that still have endogenous wildtype and presumably membrane binding copies. It could be both loss of membrane AND gain of cytoplasmic Scribble that is tumorigenic. The way it is stated in the manuscript over interprets a bit.

The interaction between TMIGD1 and Scribble was weak in the HEK cell pull-down experiments (Fig 1C). This suggests that the vast majority of Scribble does not endogenously interact with TMIGD1 (Lys only lanes have lots of Scribble). The authors present experiments from TAK-243 treated cells that artificially increase the cellular expression of both proteins by preventing their degradation. The concern is that although the interaction between the two proteins is strong *in vitro*, they don't interact with each other in cells endogenously. Work identifying TMIGD1 expressed in only a few cells and in apical membranes suggests that Scribble does not interact with TMIGD1 normally. Can the authors speculate as to why this is the case? If they use Scribble as bait in the experiments do they pull down more TMIGD1?

Data interpretation on line 220-222 is misleading. Although the two proteins bind, stating that it is the first adhesion receptor interaction without colocalization endogenously is problematic. Similar to comment 4 above. The two proteins may interact, that may be a strong but transient or random interaction which could be limited by the membranes that each protein resides on/in. Or the

interaction in vitro and in pull downs might be the result of overlap of the PBM with binding for Scribble. Can the authors show that these two proteins are found on similar membranes at endogenous levels?

CAAX domains target proteins to plasma and endo-membranes. The authors assert that expression of CAAX-PDZ-Scribble recruits TMIGD1 to the endomembrane. Can the authors show this using endocytic markers? Or organelle specific labeling?

Labels mixed up between text and panels in figure 5.

The data seem to indicate that TMIGD1 is required for lumen formation and cell polarity in MDCK cells. Some MDCK cysts expressing the dominant active TMIGD1 form a single lumen (Fig 6 graph). Are these cells polarized with polarity proteins correctly labeling apical and basal membranes? Since single lumen cysts formed in the dominant negative form of TMIGD1 experiments, can the authors speculate why this might be? Due to development? Is TMIGD1 required early in cyst opening, but not late? In other words, is it developmental?

COMMSBIO-23-0672

The cell adhesion receptor TMIGD1 recruits Scribble to the basolateral membrane via direct interaction

Eva-Maria Thüring, Christian Hartmann, Janesha C. Maddumage, Airah Javorsky, Birgitta E. Michels, Volker Gerke, Lawrence Banks, Patrick O. Humbert, Marc Kvensakul, Klaus Ebnet

Point-by-Point Response to the Reviewer's Comments

Reviewer #1 (Comments to the Author):

In the manuscript submitted by Thüring et al., the authors identified a previously unknown interaction between Scrib and a plasma membrane protein called TMIGD1. They highlight a mechanism of Scrib recruitment at the plasma membrane through its PDZ domain. The authors dissect biochemically the association between Scribble and TMIGD1 and demonstrate that this interaction is mediated by the Scrib PDZ domains, PDZ1 being the most prominent interacting domain out of the 4 PDZs of Scrib. To support this finding, the authors have also generated a crystal structure of the PDZ1 domain of Scrib with the C-terminus peptide of TMIGD1 encompassing the PDZ binding motif. Using transfection assays, they found that the recruitment of Scrib deleted of its LRR domains to the plasma membrane can be triggered by TMIGD1 full length but not by TMIGD1 devoided of its PDZ binding motif. As both Scrib and TMIGD1 have a tumor suppressing role in cancer, they explore their functional interaction in a 3D cellular model. Based on the perturbed 3D morphology following expression of TMIGD1 deleted of its Scrib binding site, they conclude that this phenotype is due to the loss of Scrib-TMIGD1 interaction.

Overall the paper characterized a novel interaction between Scrib and TMIGD1; however it fails to demonstrate the direct functional relationship between these two proteins. Intriguingly, the authors mention that many epithelial cell lines have no expression of TMIGD1 (Caco2, MDCK); however Scrib is perfectly basolateral in these cells suggesting that TMIGD1 is not mandatory for Scrib membrane localization.

As mentioned in the paper, previous works have shown that the LRR and the PDZ domains of Scrib are required for membrane localization. To show a prominent role of TMIGD1 in Scrib PDZ-mediated localization, the authors have to express the full length Scrib LRR mutant and rescue its membrane localization with TMIGD1.

A: As requested by the reviewer, we have expressed the full length Scrib LRR mutant (full length Scrib containing a mutation of Pro305 present in LRR12 of Scrib, Scrb/fl_P305L) in HEK293 cells and have analyzed its membrane localization. We found that the Scrb/fl_P305L construct was localized in the cytoplasm of WT HEK293 cells but was localized at cell-cell contacts of HEK293 cells expressing TMIGD1. In HEK293 cells expressing TMIGD1/ Δ 5, Scrb/fl_P305L was localized exclusively in the cytoplasm. These observations indicated that TMIGD1 can rescue membrane localization of the LRR mutant of Scrib. As the reviewer points out, we feel that these new observations in fact point to a prominent role of TMIGD1 in Scrib localization at the plasma membrane, which has possible implications on the tumor-suppressive function of TMIGD1. We have incorporated these new findings in the revised manuscript (Fig. 6).

Fig 1A. There is no mention of the other clones, including possibly other PDZ proteins than Scrib, that were probably selected by the two hybrid screen.

A: As the reviewer speculates, we have identified several other PDZ proteins as putative interaction partners for TMIGD1. These include Synaptojanin 2-binding protein (SYNJ2BP) and the Na(+)/H(+) exchange regulatory cofactor 2 (NHERF2). In both cases, we confirmed direct and PDZ domain-mediated interactions with TMIGD1 in biochemical assays (analogous to those performed for Scrib in Fig 1B and 4C). The interactions of TMIGD1 with SYNJ2BP and NHERF2 have recently been published (Hartmann et al (2020) BMC Mol Cell Biol. 21(1):30; Hartmann et al (2022) Sci Signal. 15(751):eabm2449). To put our findings on the identification of Scrib in a broader context, we have modified the first paragraph of the Results section and now mention the previously identified binding partners of TMIGD1. Also, we quote the Review article on TMIGD1 (Thüring et al (2023) Oncogene, Apr 22. doi: 10.1038/s41388-023-02696-5) to provide the reader with a comprehensive information on TMIGD1 and its interaction partners.

In a candidate approach we have identified NHERF1 as additional binding partner for TMIGD1. Interestingly, in the case of NHERF1 the direct (and also PDZ domain-mediated) interaction requires the presence of active ezrin, which induces an “open” NHERF1 conformation in which the two NHERF1 PDZ domains are accessible for ligand binding. This provides an example for a TMIGD1-interacting protein whose accessibility for TMIGD1 is regulated by a third protein. As outlined below in more detail, we speculate that a similar mechanism could be responsible for the weak interaction of TMIGD1 with Scrib in cells.

Fig 1C. The band corresponding to endogenous Scrib is not clear in the TMIGD1 immunoprecipitation.

A: The band representing Scrib co-precipitating with TMIGD1 is in fact weak and more difficult to see in print-outs when compared to the screen. This weak band was highly reproducible and was never observed in cells transfected with the TMIGD1/ Δ 5 construct, strongly suggesting that it is specific. The most likely explanation, therefore, is that the interaction of TMIGD1 with Scrib is weak. We think that the weak band reflects a weak interaction of TMIGD1 with Scrib in cells. To provide additional evidence for the interaction of TMIGD1 with endogenous Scrib, we have performed a reverse CoIP experiment in which we analyzed endogenous Scrib immunoprecipitates for the presence of TMIGD1. We found that TMIGD1/f.l. but not TMIGD1/ Δ 5 was present in Scrib IPs. This new observation further demonstrates that endogenous Scrib interacts with TMIGD1 in cells. We have included these new data in the manuscript (Fig. 1C of the revised manuscript).

Fig 1D. The rationale of using an inhibitor of UBE1 is not clear and the authors do not explain why this experiment has been performed. The authors should have worked with endogenous proteins to demonstrate the importance of UBE1 in the processing of Scrib and TMIGD1.

A: The rationale of using the UBE1 inhibitor was our speculation that the weak interaction of TMIGD1 with Scrib in cells despite the strong interaction observed with the recombinant proteins may be due to a degradation of TMIGD1 and/or Scrib through the ubiquitin-proteasome pathway. To address this issue we aimed at increasing the protein amount in cells by ectopically expressing both proteins in cells and at the same time inhibiting proteasome-mediated degradation. We agree with the reviewer that we had not sufficiently explained the rationale of using an inhibitor of the proteasome. In the revised version of our manuscript, we explicitly mentioned that proteasomal degradation of both TMIGD1 and Scrib had been observed before and that we used the UBE1 inhibitor to prevent degradation of TMIGD1 and Scrib and to facilitate the demonstration of their interaction in cells. In addition, we provide better pictures as well as quantifications of the CoIP experiments performed in the presence of UBE1 (Fig. 1D of the revised manuscript).

Fig 5. HEK293 cells are not really epithelial cells and are not a suitable model to study cell-cell adhesion. I suggest to the authors to perform these experiments using polarized epithelial cells such as MDCK cells. Moreover, there is no characterization of the basolateral colocalization of Scrib and TMIGD1.

A: As the reviewer points out, HEK293 cells are not typical epithelial cells in terms of apico-basal polarity. However, we think they are a useful model to study cell-cell adhesion. HEK293 cells, for example, express a N-cadherin-based adhesion complex (Flannery and Bruses (2012) *Front. Synaptic Neurosci.* 4.6). Also, ectopic expression of TMIGD1 in these cells results in increased cell aggregation, which provided evidence that TMIGD1 in fact acts as an adhesion receptor (Arafa et al (2015) *Am. J. Pathol.* 185:2757; Hartmann et al (2022) *Sci. Signal.* 15:eabm2449). Since the major emphasis of our study is to demonstrate a functional interaction of TMIGD1 with Scrib in regulating Scrib's membrane localization (but not to study the role of TMIGD1 in cell-cell adhesion and apico-basal polarity) we think that HEK293 cells provide a suitable model for our purposes. We agree with the reviewer that a localization of TMIGD1 in polarized epithelial cells would be interesting. In our previous attempts in overexpressing TMIGD1 in polarized epithelial cells we have observed that ectopically expressed TMIGD1 is localized exclusively in the cytoplasm of MDCKII cells whereas the Δ D1-TMIGD1 construct was localized at the basolateral membrane (Hartmann et al (2020) *BMC Mol Cell Biol.* 21(1):30). To address the reviewer's point, we have analyzed the recruitment of the Scrib constructs in Δ D1-TMIGD1-transfected MDCKII cells. As observed for HEK293 cells, we found that the Scrib/PDZ construct was recruited to the basolateral membrane domain whereas the Scrib/LRR construct was localized exclusively in the cytoplasm. These new observations thus demonstrated that TMIGD1 can recruit Scrib to the basolateral membrane in polarized epithelial cells. Also, since these experiments phenocopied the observations in HEK293 cells, they further support the notion that HEK293 cells are a useful model to study cell-cell adhesion-based molecular interactions. We have included these new observations in the revised manuscript (Suppl. Fig. 5).

Another issue raised by the reviewer was the characterization of the basolateral localization of Scrib. We have, therefore, stained TMIGD1-transfected HEK293 cells as well as Δ D1-TMIGD1-transfected MDCKII cells grown on polycarbonate filters for endogenous Scrib. We observed that in both cell types Scrib was localized at the basolateral membrane domain where it partially co-localized with TMIGD1. We have included these observations in the revised manuscript (Suppl. Fig. 2, Suppl. Fig. 5).

Fig 6. The results are interesting, however as TMIGD1 has probably several PDZ interactors, it does not demonstrate the direct functional relationship between Scrib and TMIGD1. Expression of a TMIGD1 mutant with a specific mutation in its C-terminus (based on the structure and lying outside of the last four residues of the PDZ binding site) abrogating the Scrib interaction could add some value to the assay.

A: As pointed out in detail in our response to point 2 of this reviewer, the hitherto described PDZ interactors include SYNJ2BP, NHERF1 and NHERF2. The interactions with these protein is abrogated upon deletion of the C-terminal PDZ binding motif (PBM, -SETAL). The crystal structure of the TMIGD1-interacting PDZ domains of these proteins in complex with the TMIGD1 peptides have not been described, yet. As a consequence, it is presently impossible to design a TMIGD1 mutation that abrogates the interaction with Scrib without affecting the interaction with the other partners. However, to test if the cyst phenotype observed upon expression of the Δ D1-TMIGD1 mutant involves the interaction with a PDZ protein, we have performed cyst assays in cells expressing a Δ D1-TMIGD1 construct that lacks the PBM (Δ D1-TMIGD1/ Δ 5). These cells predominantly developed multiluminal cysts with normal apical-basal polarity. These findings strongly suggest that apical-basal polarity formation requires TMIGD1 interaction with a PDZ domain protein. We realize that these findings do not prove that a lack of interaction with Scrib is responsible for multicellular aggregate formation in Δ D1-

TMIGD1-expressing cells. However, we feel that they further support the notion of a functional interaction of TMIGD1 with Scrib. We have incorporated these observations in the revised manuscript (Fig. 7B).

Reviewer #2 (Comments to the Author):

This paper investigates the interaction between TMIGD1 and Scribble (Scrib), proteins involved in cell adhesion and polarity. The authors show, using yeast-two hybrid and GST-pulldown experiments, that TMIGD1 directly interacts with the first PDZ domain of Scrib, and this interaction is dependent on the C-terminal PBM of TMIGD1. Co-immunoprecipitation experiments indicate that Scrib is present in TMIGD1 full-length precipitates but not in TMIGD1/ Δ 5 precipitates, confirming that the interaction is PBM-dependent in cells. The authors also demonstrate that both TMIGD1 and Scrib are subject to proteasomal degradation in HEK293 cells, and that the interaction is enhanced by the inhibitor of Ubiquitin-activating enzyme E1. Scrib mutant constructs show that the interaction is specifically dependent on the PDZ domain region of Scrib, that the 1st, 2nd, and 3rd domains have affinity, and that TMIGD1 does not interact with other LAP family members. The authors determine a crystal structure of the Scrib PDZ1 domain in complex with the C-terminal region of TMIGD1, showing the interaction is mediated through a canonical PDZ interaction. Finally, they observe Scrib localization in cells to conclude that TMIGD1 can recruit Scrib to the cell membrane in a PBM dependent manner, and that Scrib can correspondingly influence TMIGD1 localization. Overall, the paper provides some insight into a newly identified interaction between TMIGD1 and Scrib and considers the implications for cell polarity and adhesion. The crystal structure and in vitro biochemistry experiments are well performed and support the conclusions. Conversely, the imaging experiments open several questions that would need to be answered before publication. Specific concerns are:

1. The low affinity of the PDZ1-TMIGD1 complex suggests this PBM likely interacts with other PDZ domains, confirmed by its binding to 3/4 Scrib domains. Were there other PDZs found in the Y2H? This would help answer questions of specificity.

A: As suggested by this reviewer, TMIGD1 in fact interacts with other PDZ domains as well. In our Y2H screen we identified SYNJ2BP and NHERF2 as additional interactors, and we have previously shown that TMIGD1 interacts with the PDZ domain of SYNJ2BP (Hartmann et al (2020) BMC Mol Cell Biol. 21(1):30) as well as with the two PDZ domains of NHERF2 and its paralogue NHERF1 (Hartmann et al (2022) Sci Signal. 15(751):eabm2449) in a PBM-dependent manner. Notably, the interaction of TMIGD1 with NHERF1 requires active, Thr567-phosphorylated ezrin, providing an example of allosteric regulation of a TMIGD1 interaction (Hartmann et al (2022) Sci Signal. 15(751):eabm2449). In the Results section of the revised manuscript, we more explicitly mention the previously identified PDZ domain binding partners. In addition, we also quote a recently published review article on TMIGD1 which describes all interaction partners that have been identified so far (Thüring et al (2023) Oncogene, Apr 22. doi: 10.1038/s41388-023-02696-5).

2. The interaction between recombinant TMIGD1 and endogenous Scrib is extremely weak, despite the PBM protein being expressed at high levels (also true with recombinant full length Scrib in Fig 2). As there are many known binding partners of the Scrib PDZ domains it is important to contrast the TMIGD1 interaction with others. Indeed, this is the weakest element of the manuscript – there was no attempt to inform us of all known PDZ1/PDZ3 interactions and contrast the TMIGD1 complex with alternatives. Do co-IPs of endogenous Scrib with other partners containing PBMs result in a more

robust interaction?

A: We agree with the reviewer that the interaction between recombinant TMIGD1 with Scrib is very weak. We have previously made a similar observation for the interaction between the TMIGD1-related cell adhesion receptor JAM-A and the polarity protein PAR-3, i.e. a very strong interaction between recombinant proteins *in vitro* but a very weak interaction between endogenous proteins in cells (Ebnet et al (2001) EMBO J. 20:3738). We explain this by the nature of the scaffolding proteins which dynamically interact with many other proteins in cells, and depending on the spatial and temporal context bind to a given partner protein with different efficiencies. In the Discussion of our original manuscript, we have discussed various possibilities as to why the interaction between TMIGD1 and Scrib in cells may be weak. We refrained from providing a comprehensive overview of all Scrib PDZ1/PDZ3-interacting proteins since we think this would rather be the topic of a review article. Instead, we have quoted several review articles in which the PDZ interactors are comprehensively summarized (Refs 15, 16, 17). Along the same line, we refrained from picking a single interaction and studying the strength of interaction by CoIP, as the robustness of an interaction most likely varies between different binding partners and may also depend on the cellular context. However, to put the interaction of TMIGD1 with the Scrib PDZ domains 1 and 3 in a context with known binding partners of Scrib, we provide a table in which we depict the affinities of the Scrib PDZ1/3 – TMIGD1 interactions in comparison with the affinities of Scrib PDZ1/3 interactions with several known ligands, including adenomatous polyposis coli (APC), PAK-interacting exchange factor beta (β -PIX), mutated in colorectal cancer (MCC), Vang-like protein 2 (Vangl2) and the HTLV-1 Tax1 protein. This table shows that the affinities of the TMIGD1 – Scrib interactions are in very similar range like those of these ligands. We have included this table in the revised version of the manuscript (Table 3).

3. The conclusions of the TAK-243 inhibitor experiment were that both proteins are modulated by ubiquitination/degradation. The levels in treated/untreated do not look significantly different to me, and to make this conclusion requires quantitation.

A: As requested by the reviewer, we provide quantitations of the CoIP data. In addition, we provide Western blot images which better reflect the increase in expression levels of the two proteins and the increase in the amount of Scrib co-precipitated with TMIGD1 after treatment with TAK-243. The quantitations are part of Fig. 1D of the revised manuscript.

4. The inclusion of Scrib/PDZ-Caax was not well rationalized. What was the purpose of this experiment?

A: Based on this reviewer's comment and also on a similar comment of reviewer #3, we realized that we had not sufficiently elaborated on the rationale of the Scrib/CAAX experiment in the original manuscript. We performed the CAAX experiments to obtain further evidence for the interaction of TMIGD1 with Scrib. Since we could not exclude the possibility that the Scrib/PDZ construct does not interact with TMIGD1 due to cytoplasmic localization, we envisaged that we could force membrane localization of Scrib/PDZ by addition of CAAX prenylation motif. The addition of the CAAX prenylation motif did not alter the interaction of the Scrib/PDZ construct (Fig. 2) but resulted in an ectopic recruitment of TMIGD1 to endomembranes (Fig. 5B). We considered this observation as additional evidence that TMIGD1 and Scrib interact in cells without claiming a functional relevance for this ectopic localization of TMIGD1. In the revised version of the manuscript we have more explicitly explained why this experiment was performed (Results section of the revised manuscript). In addition, as requested by reviewer #3, we provide evidence that Scrib/PDZ-CAAX and TMIGD1 co-localize at the trans-Golgi network (TGN) which suggests that TGN-localized Scrib-CAAX prevents normal trafficking of TMIGD1 to the surface. We have included these findings in the revised manuscript (Suppl. Fig. 4).

5. 2 alternative LAP proteins were assayed for TMIGD1 interaction, but the authors did not inform us of how many proteins are in the LAP family or how representative these two members are of the LAP family as a whole.

A: As requested by the reviewer, we provide more detailed information on the LAP family proteins and their partially redundant functions in polarized epithelial cells. We feel that it has now become more clear why we addressed the possibility that TMIGD1 interacts with the two other LAP family members expressed in epithelial cells, i.e. Erbin and Lano. This additional information is included in the Results section of the revised manuscript.

6. The crystal structure is high resolution and was well refined. To place it in context with previous work, it would be informative to compare the major PDZ1 binding determinants with other PDZ1-peptide complex structures.

A: As suggested by the reviewer, we have included a table in which we compare the direct contacts (hydrogen bonds, salt bridges) between TMIGD1 and Scrib with direct contacts between other ligands (APC, β -PIX, Vangl2) and Scrib/PDZ1. This table is included as Table 1 of the revised manuscript. In addition, we have included a table in which we compare the affinities of TMIGD1 – Scrib/PDZ1 interaction with the affinities of other Scrib PDZ1 binding partners including APC, β -PIX, MCC, Human T-cell leukemia virus 1 (HTLV-1) protein Tax1, and Vangl2. We also included a comparison of the TMIGD1 – Scrib/PDZ3 interaction with the same ligands. As can be seen in this table, the affinities of the TMIGD1 – Scrib/PDZ interactions are in a similar range as the affinities of Scrib/PDZ interaction with other ligands. This table is included as Table 3 of the revised manuscript.

7. How much GST proteins were used for the pull downs in Fig 4b? Seems like very small amounts to get such a robust binding when all previous interactions were so weak.

A: As described in the Methods section, we routinely used 3 μ g of GST proteins per sample in the pulldown experiments (Fig. 1B, Fig. 4B). The Coomassie Brilliant Blue stainings shown in the bottom part of the Fig. 4B were performed with 0.3 μ g of GST protein (10% of input). To avoid confusion, we provide the information on the amount of GST fusion proteins analyzed in the CBB stainings in the legend to the figure (Legends to Fig. 1B and Fig. 4B). As an additional note to the reviewer, we also observed a strong interaction in all in vitro binding experiments with recombinant proteins (Fig. 1B, Fig. 4B). We think that the most likely explanation for the discrepancy between very strong in vitro pulldown experiments and weak interactions in CoIP experiments is the presence of other interaction partners in cells which might bind and influence the interaction with TMIGD1.

8. The rationale for observing loss of TMIGD1 binding with the PDZ2 mutant but not detecting an in vitro interaction (ITC) is not clear to me. The 4 PDZ domains of Scrib have been extensively studied, and there is no known intramolecular regulation of the PDZ2 domain. The paper cited as evidence of cooperation (alpha1D-adrenergic receptor) uses BLI and is not truly measuring cooperative binding, but rather avidity of multiple domains with overlapping specificity. How do the authors envision a cooperative mechanism impacting PDZ2 affinity for TMIGD1 in the pull down experiment but they observe no in vitro binding of the domain alone?

A: In the revised version of our manuscript (Discussion), we discuss in more detail a possible mechanism that could explain the discrepancy between our in vitro binding experiments and the GST pulldown experiments. Specifically, we discuss the possibility that the Scrib PDZ1-4 module could form a supramodule in which PDZ occupation or regions adjacent to PDZ domains could influence

accessibility of other PDZ domains. Similar mechanisms have been described for other PDZ domain proteins including Scrib itself (Ren et al (2015) *Biochem. J.* 468:133) and PSD-95 (Rademacher et al (2019) *eLife* Mar 13;8:e41299; Laursen et al (2020) *PNAS* 117:24294). However, we have no clear-cut explanation for our observation that the inactivation of PDZ2 reduces TMIGD1 binding although the PDZ2 in isolation does not interact with the TMIGD1 peptide, and we emphasize this in the discussion.

9. Was the Scrib PDZ3 picked up by Y2H? If not, was there a construct in the cDNA library that encompassed this region?

A: We did not pick up PDZ3 of Scrib in the Y2H screen. We performed two Y2H screens and picked up a total number of 20 (6 + 14) partially overlapping clones, which all encompassed a total region of AA633 – 836 of Scribble. The reason why we did not pick up PDZ3 is not clear. Since Y2H experiments are highly sensitive we think the most likely explanation is that a clone containing PDZ3 was either not present or underrepresented in the library, as the reviewer speculates. We have previously made a similar observation for the cell adhesion molecule JAM-A. In a Y2H experiment using the same Y2H library we identified Afadin and PAR-3 as binding partners but not ZO-1. The latter was identified in candidate approaches in biochemical experiments (Ebnet et al (2000) *JBC* 275:27979; Ebnet et al. (2001) *EMBO J.* 20:3738).

10. More effort should be made to place the findings in context with the extensive Scrib PDZ literature. How many PDZ1 ligands are known and what are their affinities? How does the TMIGD1 affinity compare?

A: The literature on Scrib is in fact extensive. In the Discussion section of our manuscript, we have mentioned that at least 37 proteins are described to directly interact with Scrib, and we have quoted a review article which provides a summary of these interaction partners. Unfortunately, for the majority of Scrib ligands detailed studies on the affinities of the interactions are not available. We agree with the reviewer that a direct comparison of affinities would be helpful to place the TMIGD1 interaction in a broader context. In the revised version of the manuscript, we have included a table in which we directly compare the affinities of five other Scrib ligands for PDZ1 and PDZ3 with those of TMIGD1. As can be seen in this direct comparison, the affinity of TMIGD1 with Scrib PDZ1 and PDZ3 is within the range described for the other ligands. We think that this table will help the reader to put the TMIGD1 interaction with Scrib in a broader context. This table is included as Table 3 in the revised manuscript.

11. The imaging experiments are the weakest component of the manuscript and should be improved. First, the LRR of Scrib has long been considered the membrane recruitment region of this protein (as discussed extensively by the authors in the discussion). Why does the Scrib LRR here not localize to the membrane? For TMIGD1, the protein has a transmembrane domain but why does deletion of the final 5 residues of the cytoplasmic domain alter its localization? Should it not remain on the membrane? The authors state that “TMIGD1 as an adhesion receptor localized at cell-cell contacts of polarized epithelial cells which recruits Scrib to the lateral membrane domain” but there is no evidence of either in this manuscript. I see no evidence that TMIGD1 is at cell-cell contacts, only that it can be membrane localized. In polarized epithelial cells does it colocalize with markers of tight or adherens junctions? It appears more apical in the cysts presented here. And for evidence that Scrib is recruited to the lateral membrane we would need to see confocal data, z stacks, that this is true. The HEK images seem to show it is membrane localized. Finally, how is it that full length Scrib in these cells is at the membrane, but the LRR and PDZ regions alone are not. What is major recruitment factor for Scrib membrane localization in HEK cells?

A: The reviewer raises several important questions which are central to the understanding of Scrib membrane localization and, thus, of Scrib's tumor-suppressive function but also for the understanding of TMIGD1 and its tumor-suppressive function. Given the complexity of the Scrib membrane recruitment mechanisms, we can only speculate as to why the Scrib mutants are not recruited to cell-cell contacts.

As to the localization of Scrib constructs in these cells, we think that a possible explanation for the lack of Scrib/LRR membrane localization is that the membrane recruitment mechanisms for the LRR region alone is not operative in HEK293 cells. In fact, only 4 binding partners were identified for the LRR region, whereas 37 PDZ domain binding partners are described (Bonello and Peifer (2019) *JCB* 218:742). Obviously, not all interaction partners are expressed by a given cell type, and it is thus possible that the LRR ligands are not present in HEK293 cells. As to the lack of recruitment of the Scrib/PDZ construct, we speculate that a functional LRR region is required for the PDZ domain-mediated membrane recruitment of Scrib. This would at least explain our observation that Scrib/f.l. but not Scrib/PDZ is recruited by endogenous mechanisms. Our new observations using a full length Scrib construct with a mutation in the Pro305 residue (Scrib/P305L) support this view. For the revised version of our manuscript we have performed recruitment experiments with Scrib/P305L. We observed that Scrib/P305L does not localize to cell-cell contacts in HEK293/WT cells but is prominently present in HEK293/TMIGD1 cells. These observations further support the view that a functional LRR region is required for membrane recruitment, which is in line with published data. In addition, it further underlines a critical function for TMIGD1 as it can rescue membrane recruitment of the Scrib/P305L mutant, which may be part of its tumor-suppressive activity. As to the mechanisms of Scrib/f.l. membrane localization in HEK293 WT cells, we speculate that the cadherin/catenin complex could be responsible. HEK293 cells express N-cadherin (Flannery and Bruses (2012) *Front. Synaptic Neurosci.* 4.6), a classical cadherin that interacts with β -catenin and α -catenin. Notably, β -catenin contains a C-terminal PBM through which it directly interacts with Scrib PDZ1 and PDZ3 (Zhang et al (2006) *JBC* 281:22299; Sun et al (2009) *MBoC* 20:3390). Thus, β -catenin associated with N-cadherin would be a possible mechanism of Scrib recruitment to cell-cell contacts in HEK293 cells.

We agree with the reviewer that TMIGD1/ $\Delta 5$ is expected to localize at the membrane. We have presently no explanation why TMIGD1/ $\Delta 5$ is localized exclusively in the cytoplasm. We have observed previously for JAM-A that its recruitment to cell-cell contacts in subconfluent Caco2 cells depends on its PBM-mediated interaction with AF-6/Afadin (Ebnet et al (2000) *JBC* 275:27979) opening the possibility that TMIGD1 requires a PDZ-mediated interaction for cell surface localization, either for trafficking or for retention at the membrane. As pointed out before, TMIGD1 interacts with the two scaffolding NHERF1 and NHERF2 through its PBM (Hartmann et al (2020) *Sci Signal.* 15(751):eabm2449). Both NHERF proteins were found to regulate the trafficking and cell surface localization of the glutamate transporter GLAST in HEK293 cells (Sato et al (2013) *BBRC* 430:839). If a similar mechanism exists for the trafficking of TMIGD1, the absence of the PBM would prevent its efficient trafficking to the cell surface. The mechanisms regulating TMIGD1 trafficking and surface localization are clearly highly relevant for the understanding of its biology and will be an important aspect for our future research.

As already pointed in our response to Reviewer #1, when ectopically expressed in MDCKII cells TMIGD1/f.l. is localized exclusively in the cytoplasm whereas the $\Delta D1$ -TMIGD1 construct is localized at the basolateral membrane (Hartmann et al (2020) *BMC Mol Cell Biol.* 21(1):30). We have, therefore, used $\Delta D1$ -TMIGD1-transfected MDCKII cells as a model system for polarized epithelial cells to analyze the recruitment of Scrib constructs. We made the same observations as in HEK293 cells, i.e. efficient membrane recruitment of Scrib/PDZ and exclusive cytoplasmic localization of Scrib/LRR. These experiments thus suggested that the TMIGD1-mediated recruitment of Scrib to cell-cell contacts are relevant in polarized epithelial cells as well. We have included these new observations in the revised manuscript (Suppl. Fig. 5).

To address the reviewer's question on cell-cell contact localization of TMIGD1 in HEK293 cells or in polarized cells we have performed confocal analyses of TMIGD1 localization. In HEK293 cells we found that TMIGD1/f.l. was clearly localized at cell-cell contact sites. Similarly, in MDCKII cells the transfected $\Delta D1$ -TMIGD1 construct localized at the apical membrane domain, as observed previously

for TMIGD1 in Caco2 cells (Hartmann et al (2022) *Sci Signal.* 15(751):eabm2449), but was also very clearly localized at cell-cell contact sites. Together with our previous observations in HK-2 cells, in which TMIGD1 localizes to cell-cell contacts with increasing cell confluency (Hartmann et al (2020) *BMC Mol Cell Biol.* 21(1):30), these findings indicate that in addition to its apical localization TMIGD1 localizes to cell-cell contacts in epithelial cells. We have included the confocal stainings of TMIGD1 in HEK293 cells and of Δ D1-TMIGD1 in MDCKII cells in the revised manuscript (Suppl. Fig. 2 and Suppl. Fig. 5, respectively).

12. The PDZ1/2/3 mutants should be included in the imaging analysis to verify mutational inactivation of these modules disrupts membrane recruitment.

A: We have included imaging analyses of the three PDZ mutants in our revised manuscript. As expected from our biochemical experiments, which showed that TMIGD1 interacts with several PDZ domains of Scrib, we observed that mutations of individual PDZ domains did not affect membrane recruitment of the Scrib construct, supporting the view that several PDZ domains can mediate membrane localization in cells. We have included these new observations in the revised version of the manuscript (Suppl. Fig. 3).

13. The MDCK cyst experiments show that expression of TMIGD1 lacking the Ig domain induces aggregates and significant loss of well-formed cysts with a single lumen. This system, however, was not exploited to determine whether this has anything to do with Scrib or even with the PBM of TMIGD1. It is therefore unclear what this experiment brings to the current manuscript. Why is the Δ D1 variant of TMIGD1 used here? What happens with full length TMIGD1? Also, an experiment with the Δ PBM variant would inform if the phenotype is dependent on binding to a PDZ domain. Finally, rather than use the red channel for actin there is an opportunity to stain for endogenous Scrib, which is highly expressed in these cells. Some further experiments here are required to support the conclusions.

A: The purpose of the MDCK cyst experiment was to address the question if TMIGD1 regulates epithelial cell polarity, which was suggested by its interaction with cell polarity protein Scrib. MDCK cyst assays are a widely used system for the analysis of apico-basal polarity (Datta et al (2011) *Curr Biol.* 21:R126). A failure in developing apico-basal membrane polarity results in multicellular aggregates providing a simple functional read-out of cell polarity. The reason to use the Δ D1 variant (Δ D1-TMIGD1) instead of full length TMIGD1 was our previous observation that ectopic TMIGD1/f.l. localized in the cytoplasm of MDCKII cells whereas the Δ D1-TMIGD1 mutant was enriched at cell-cell contacts through unknown mechanisms (Hartmann et al (2020) *BMC Mol Cell Biol.* 21(1):30). On the basis of our own observations with the TMIGD1-related adhesion molecule JAM-A, in which an analogous construct acts in a dominant-negative manner in various assay systems (contact inhibition of locomotion (Kummer et al (2022) *JCB* 221(4):e202105147), mitotic spindle orientation (Tuncay et al (2015) *Nat Commun.* 6:8128), epithelial barrier function and MDCK cyst formation (Rehder et al (2006) *Exp Cell Res.* 312:3389)), we hypothesized that the Δ D1-TMIGD1 mutant might also act in a dominant-negative manner. Our observations that the Δ D1-TMIGD1 results in multicellular aggregates instead of lumen-containing cysts supported the idea that TMIGD1 is a regulator of apico-basal polarity in polarized epithelial cells. We agree with the reviewer that the MDCK cyst experiments do not allow a conclusion of a Scrib-mediated effect on lumen formation. Therefore, as requested by the reviewer, we have performed MDCK cyst experiments with cells expressing the Δ PBM variant (Δ D1-TMIGD1/ Δ 5). We observed that these cells develop lumen-containing cysts, which strongly suggests that the phenotype observed for Δ D1-TMIGD1 cells (multicellular aggregates without lumens) is due to a loss of PBM-dependent interaction, such as Scrib. Surprisingly, cysts derived from Δ D1-TMIGD1/ Δ 5 cells predominantly contained multiple lumens instead of a single lumen observed in control cells. We interpret these findings as an evidence that TMIGD1 regulates cysts development through additional

activities that are independent of PDZ interactions, possibly by its interaction with Ezrin (Rahimi et al (2021) J Biomed Sci. 9;28(1):61) which is an apical membrane-localized protein in Caco2 cysts (Michaux et al (2016) Biol Cell 8:19-28). As to the reviewers notion concerning stainings for Scrib in MDCK cysts, we did not observe any changes in Scrib membrane localization in Δ D1-expressing cells, which we think is not surprising as the Δ D1-TMIGD1 construct localizes at the basolateral membrane domain in these cells (Suppl. Fig. 5). We are aware of the fact that we cannot provide a mechanism that explains the cyst phenotype. However, we feel that the MDCK cyst experiments provide a first step in the understanding of the function of TMIGD1 in the context of cell polarity. In the revised version of the manuscript, we have more clearly explained the purpose of the MDCK cyst experiment (Results section). Furthermore, we have included our new observations on cyst formation in MDCKII cells expressing the Δ PBM variant (Fig. 7B of the revised manuscript).

14. Finally, relevance of TMIGD1 recruitment of Scrib is not completely clear to me. TMIGD1 is expressed only in gastrointestinal cells and even in these cells is difficult to detect at the protein level. The authors need to clarify how important they believe TMIGD1 is to normal Scrib function – or do they contend this interaction is relevant only in cancer cells? How do they place it in context of all the other Scrib PDZ binding partners? The designation of TMIGD1 as an ‘adhesion receptor’ localized as cell-cell contacts needs further validation as well, as I could not find strong data indicating this in the literature.

A: As the reviewer correctly points out, the physiological relevance of the Scrib recruitment by TMIGD1 is not clear at this stage. The purpose of our study is to characterize the interaction of TMIGD1 with Scrib, and we are aware of the fact that several subsequent studies are necessary to understand the relevance of this interaction in cells and tissues. As we have eluded to in some detail in our manuscript (Introduction, Discussion), TMIGD1 is expressed not only in gastrointestinal cells but also in kidney cells (Refs 21, 22, 23 of our manuscript), in particular in epithelial cells lining the proximal tubules where it is enriched at cell-cell junctions (Ref 23 of our manuscript). Thus, it is possible that TMIGD1 localized at cell-cell contacts of polarized epithelial cells lining the proximal kidney tubules contributes to Scrib membrane localization to support Scrib’s function as tumor suppressor and regulator of membrane polarity. As to the question on how we place the interaction of Scrib with TMIGD1 into a context of all other binding partners, we believe that – given the large number of Scrib interactions (45 binary interactions acc. Uniprot; <https://www.uniprot.org/uniprotkb/Q14160/entry>) – it would go far beyond the scope of our manuscript to discuss all other Scrib binding partners. As mentioned above, the major purpose of our study is to describe and characterize the interaction between the two tumor suppressor proteins TMIGD1 and Scribble. As detailed in the last paragraph of the Discussion, we think that a loss of TMIGD1 expression in intestinal or kidney epithelial cells – together with other mutations – could contribute to a loss of Scrib membrane localization. As to the designation of TMIGD1 as an “adhesion receptor”, we and others have in fact used several approaches to characterize the property of TMIGD1. First, ectopic expression of TMIGD1 in HEK293 cells results in cellular aggregation (Arafa et al (2015) Am J Pathol. 185:2757; Hartmann et al (2022) Sci Signal. 15(751):eabm2449); second, endogenously expressed TMIGD1 co-immunoprecipitates with ectopically expressed Flag-TMIGD1 (Hartmann et al (2022) Sci Signal. 15(751):eabm2449), and, similarly, GST-TMIGD1 can pulldown ectopically expressed TMIGD1 from transfected HEK293 cells (Arafa et al (2015) Am J Pathol. 185:2757); third, coating fluorescently labelled beads with a TMIGD1 fusion protein consisting of the extracellular domain of TMIGD1 fused to the Fc part of human IgG results in bead aggregation (Hartmann et al (2022) Sci Signal. 15(751):eabm2449). We believe that these observations provide a strong evidence for an adhesive function of TMIGD1 to justify its designation as adhesion receptor.

Reviewer #3 (Comments to the Author):

Thuring et al., present a manuscript that follows up on a hit from a yeast two hybrid protein-protein interaction between the cell polarity protein Scribble and the cell adhesion molecule TMIGD1. They first validate the yeast two-hybrid result by showing that TMIGD1 interacts with Scribble in vitro before showing they weakly interact when expressed at high levels in cells (through preventing their degradation). They subsequently present data showing that the interaction is dependent on a PDZ binding motif (PBM) at the carboxy terminal of TMIGD1 with the PDZ portion specifically of Scribble, and not its relatives Erbin or Lano. Crystal structure analysis of the TMIGD1 PBM in complex with Scribble PDZ1 shows amino acid specific interaction in the PDZ binding pocket. Further in vitro binding assays show that TMIGD1 binds to PDZ1 and PDZ3 of Scribble. Lastly, Thuring and colleagues transiently express tagged versions of the TMIGD1 and Scribble in HEK293 and MDCK cells to understand in vivo dynamics of Scribble membrane recruitment and the role of TMIGD1 in cell polarity and luminogenesis.

Overall, the manuscript presents an intriguing finding, that the cell adhesion molecule TMIGD1 binds to Scribble PDZ domains 1 and 3. The authors assert this novel interaction between the two proteins forms the basis of Scribble membrane recruitment. They provide strong evidence from in vitro and gain of function in vivo protein interaction studies that couple well with atomic resolution data for the interaction. However, the in vivo studies are lacking the necessary evidence to make the bold claim that this interaction is sufficient to explain how Scribble is recruited to the membrane in normal cells or that lack of this interaction is a driving cause of tumorigenesis. Below are the major concerns the authors should consider addressing before publication. The bulk of the concerns deal with over interpretation of published work and gathered data.

Comments for major concerns:

The title of the manuscript is misleading and over-interprets the data presented. Although the data show that Scribble can interact with TMIGD1, its relevance in a wide range of normal cells and tissues, or the lack of interaction in disease, is not deeply explored (see below). Data from gain of function experiments show that certain forms of truncated Scribble can be recruited to the membrane. A title change that better reflects the data presented is necessary.

A: We agree with the reviewer that our findings on Scrib membrane recruitment through a direct interaction with TMIGD1 slightly overinterpreted our data because we cannot prove that the recruitment in cells is mediated through a direct interaction and also because the data were based on truncated Scrib constructs. It is generally very difficult to prove a direct interaction of two proteins in a cell. We think that our biochemical and structural evidence on a direct interaction of TMIGD1 with Scrib makes a direct interaction in cells very likely. To avoid any overinterpretation of our data, and on the basis of new observations, we have changed the title (see below). In the revised version of our manuscript, we have added additional observations with a Scrib full length construct that contains the P305L mutation. While this mutant is not recruited to the membrane in normal HEK293 cells, it is recruited to the membrane in TMIGD1- but not TMIGD1/ Δ 5- expressing HEK293 cells. We feel that these new findings add more physiological relevance to our findings, in particular because the Scrib/P305L mutant has been shown to result in tumor formation (Feigin et al (2014) Cancer Res. 74:3180). Taking these new observations into account, we have change the title to "Membrane recruitment of the polarity protein Scribble by the cell adhesion receptor TMIGD1". We think that this new title does not overinterpret our findings and thus adequately reflects the observations made in our study.

The authors make the claim that the mechanism of Scribble membrane localization is not well understood (see lines 40-4, 92-93). This is not true. The full molecular mechanism of Scribble membrane localization may not be entirely understood especially as there may be mechanisms that vary between organisms, cell-types, and other aspects of biological context. The authors cite the Bonello and Peifer review with a whole section (Fig 2) which contains multiple examples of the known mechanisms for Scribble membrane interaction. The bold claim that membrane localization is unclear is not true. Binding to TMIGD1 may be another mechanism that is certainly context specific (since TMIGD1 is only expressed in a few cell types). The authors should reframe the overarching rationale. Finding the binding partners of Scribble is in itself interesting. They should also revisit lines 98-99, which makes the assumption that it is a cell adhesion molecule that Scribble requires. Why an adhesion molecule versus any other lipid molecule, transmembrane protein or cortex binding protein?

A: Based on the reviewer's remarks we realized that our notion on Scrib's membrane localization and TMIGD1 as membrane anchor of Scrib might be misinterpreted as to claim that TMIGD1 is the only membrane anchor in cells. We have, therefore, carefully revised our manuscript to avoid the impression that the mechanisms underlying Scrib membrane localization are not well understood. Instead, where applicable, we claim that these mechanisms are not fully understood. In addition, we avoid the impression that a cell adhesion molecule is mandatory for Scrib membrane localization. We have rephrased the Introduction (lines 98-99) accordingly.

The assertion that it is the membrane localization of Scribble that prevents tumors, versus its cytoplasmic localization that promotes tumors, is not well reasoned. Statements on lines 39-40 and 89-91 reframe the original findings from Refs 17-19 that tumors resulting from cytoplasmic localization of Scribble mean that membrane localization is required. It is possibly true that the original finding of tumors that result from the P305L mutation has nothing to do with the membrane localization but rather sequestration of tumorigenic factors in the cytoplasm. That idea being that the cytoplasm localization could also be tumorigenic. The authors should reframe their argument to fit the data from the references they cite. In fact those papers also over-express membrane miss-localization versions of Scribble, that still have endogenous wildtype and presumably membrane binding copies. It could be both loss of membrane AND gain of cytoplasmic Scribble that is tumorigenic. The way it is stated in the manuscript over interprets a bit.

A: Based on this comment we realized that our statements on the cell membrane and cytosolic localization of Scrib and the role of Scrib as tumor suppressor and tumor promoter suggested a causal relation between localization and tumor-suppressive/tumorigenic function. To avoid this impression, we have rephrased all relevant text passages in our manuscript. Specifically, we mention that its subcellular localization correlates with its tumorsuppressive/tumorigenic functions, and we also mention that the data shown in the cited references (refs 17-19) are suggestive rather than indicative for a tumor-promoting function of cytosolic Scrib.

The interaction between TMIGD1 and Scribble was weak in the HEK cell pull-down experiments (Fig 1C). This suggests that the vast majority of Scribble does not endogenously interact with TMIGD1 (Lys only lanes have lots of Scribble). The authors present experiments from TAK-243 treated cells that artificially increase the cellular expression of both proteins by preventing their degradation. The concern is that although the interaction between the two proteins is strong in vitro, they don't interact with each other in cells endogenously. Work identifying TMIGD1 expressed in only a few cells and in apical membranes suggests that Scribble does not interact with TMIGD1 normally. Can the authors speculate as to why this is the case? If they use Scribble as bait in the experiments do they pull down more TMIGD1?

A: This issue has also been raised by the other reviewers. Given the strong interaction of recombinant proteins in vitro, the weak interaction in cells is somewhat surprising. As this reviewer points out, we think that one likely explanation is that the majority of Scrib is not associated with TMIGD1. Given the numerous interaction partners described for Scrib, endogenous Scrib could be outcompeted from TMIGD1 binding. One additional possible explanation is that both proteins are subject to proteasomal degradation. We have included quantifications of our TAK-243 experiments which show an approx. 4-fold increase in the amount of Scrib co-precipitating with TMIGD1 (Fig. 1D of the revised manuscript). As to our previous findings on TMIGD1 localization at the apical membrane, we feel that the apical TMIGD1 localization does not exclude the possibility that TMIGD1 interacts with other proteins like Scrib at the basolateral membrane. In fact, we found that in transfected MDCKII cells, a TMIGD1 construct that lacks the membrane-distal Ig domain (Δ D1-TMIGD1) localizes to both the apical membrane and the basolateral membrane (see also next comment of this reviewer, data is shown in Suppl. Fig. 5 of the revised manuscript). We speculate that – given the promiscuous nature of Scrib – its relative localization in cells is influenced by many parameters including the specific affinities towards its interactors, and that the presence or absence of binding partners at one particular location influences its interaction with other binding partners. To obtain additional evidence for a TMIGD1 – Scrib interaction in cells, we have performed a reverse CoIP experiment in which we used Scrib as bait, as requested by this reviewer. We observed a slightly stronger interaction when we used Scrib as bait to pulldown TMIGD1. We consider this result as an additional evidence that TMIGD1 and Scrib interact in cells. We have included these new data in the revised manuscript (Fig. 1C).

Data interpretation on line 220-222 is misleading. Although the two proteins bind, stating that it is the first adhesion receptor interaction without colocalization endogenously is problematic. Similar to comment 4 above. The two proteins may interact, that may be a strong but transient or random interaction which could be limited by the membranes that each protein resides on/in. Or the interaction in vitro and in pull downs might be the result of overlap of the PBM with binding for Scribble. Can the authors show that these two proteins are found on similar membranes at endogenous levels?

A: We think that our interpretation of TMIGD1 being the first hitherto identified adhesion receptor with the potential to recruit Scrib to the basolateral membrane is not an overinterpretation of the data. We have found previously that TMIGD1 localizes to the basolateral membrane domain in human kidney epithelial cells (Hartmann et al (2020) BMC Mol Cell Biol. 21(1):30). To further support our interpretation, we provide confocal sections of TMIGD1-transfected HEK293 cells stained for ectopic TMIGD1 and endogenous Scrib (Suppl. Fig. 2 of the revised manuscript). In addition, we provide confocal sections of Δ D1-TMIGD1-transfected MDCKII cells stained for ectopic TMIGD1 and endogenous Scrib (Suppl. Fig. 5A of the revised manuscript). Both stainings show that endogenous Scrib localizes the lateral membrane domain where it partially co-localizes with TMIGD1. We think that these findings strongly support our notion of TMIGD1 serving as epithelial adhesion receptor that can recruit Scrib to cell-cell contact sites.

CAAX domains target proteins to plasma and endo-membranes. The authors assert that expression of CAAX-PDZ-Scribble recruits TMIGD1 to the endomembrane. Can the authors show this using endocytic markers? Or organelle specific labeling?

A: Based on this comment and on a comment of Reviewer #2 we realized that we had not adequately explained the rationale behind the CAAX experiment. We performed the CAAX experiment to obtain additional evidence for an interaction between TMIGD1 and Scrib in cells. We hypothesized that the CAAX motif added to the Scrib/PDZ construct may result in its ectopic localization at endomembranes such as the ER or Golgi (Ref 40), and that an ectopic recruitment of TMIGD1 to endomembranes would thus provide additional evidence for an interaction between the two proteins. Since this was in fact

the case, we think that these observations provide additional evidence for a mutual interaction. In our revised manuscript, we avoided any statement to say that the co-localization of the ectopic proteins at endomembranes is of physiological relevance. To avoid any misconception, we have more explicitly explained why this experiment was performed (Results section of the revised manuscript). Finally, to identify the compartment in which Scrib/PDZ-CAAX and TMIGD1 co-localize we have performed co-stainings with markers for the Golgi apparatus (TGN46) and the endoplasmic reticulum (KDEL). We observed that Scrib-CAAX co-localized with TGN46 but not with KDEL, strongly suggesting that Golgi-localized Scrib-CAAX hinders the normal trafficking of TMIGD1 to the cell surface. We think that these findings provide a reasonable explanation for the co-localization of TMIGD1 with Scrib-CAAX. We have incorporated these observations in our revised manuscript (Suppl. Fig. S4).

Labels mixed up between text and panels in figure 5.

A: The labeling error has been corrected.

The data seem to indicate that TMIGD1 is required for lumen formation and cell polarity in MDCK cells. Some MDCK cysts expressing the dominant active TMIGD1 form a single lumen (Fig 6 graph). Are these cells polarized with polarity proteins correctly labeling apical and basal membranes? Since single lumen cysts formed in the dominant negative form of TMIGD1 experiments, can the authors speculate why this might be? Due to development? Is TMIGD1 required early in cyst opening, but not late? In other words, is it developmental?

A: As the reviewer points out, some MDCK cysts with induced expression of the dominant active TMIGD1 construct form a single lumen. Apico-basal polarity was normal in these cysts as indicated by a strong cortical F-actin signal at the apical membrane of the cells. We have made similar observations in MDCK cysts expressing dominant active forms of JAM-A (Δ V-JAM-A or JAM-A/MC, approx. 40% of single lumen-containing cysts; Rehder et al (2006) *Exp Cell Res* 312:3389). We, therefore, think that this phenotype is not specifically related to the activity of dominant active TMIGD1. Our stainings for F-actin, which on the basis of the terminal web strongly labels the apical membrane domain and thus can be used as a read-out for polarization, indicates that the cells are normally polarized. We have no clear-cut explanation as to why a significant number of cysts is normal upon expression of dominant active TMIGD1 (or dominant active JAM-A). One possibility would be heterogeneities in the expression levels of the dominant active proteins, and that a lower expression levels of the dominant active proteins are not sufficient to induce the cyst phenotype.

Reviewers' comments:

Reviewer #1 (Remarks to the Author):

The authors have satisfactorily addressed most of my concerns.

Reviewer #2 (Remarks to the Author):

The revised manuscript from Thüring et al. includes the following additional data:

- Fig 1c: reciprocal co-IP
- Fig 1d: quantitation of SCRIB levels following TAK243 addition
- Fig 6: SCRIB P305L localization with TMIGD1 fl/ Δ 5
- Fig 7b: cyst formation with TMIGD1 Δ 5
- Supp Fig 2: confocal images of HEK cells showing z stacks (β -CAT and SCRIB)
- Supp Fig 3: images in HEK cells of individual SCRIB PDZ mutants
- Supp Fig 4: SCRIB PDZ-Caax at Golgi (not ER)
- Supp Fig 5: SCRIB in MDCK cells described as "TMIGD1 recruits Scrib in polarized epithelial cells" along with MDCK expression of SCRIB LRR or PDZ

The authors have made an attempt to address the many concerns raised by the reviewers of the first submission. The major concerns were the weakness of the interaction in cells (compared to in vitro), the specificity of the interaction, a lack of clarity for the rationale of some experiments, the choice of 293 cells as a model for cell-cell adhesion and polarity, improved attempts to compare the interaction with other PDZ domain binders, and to derive more functional relevance from some of the experiments – particularly the cyst assay. Moreover, the reviewers were concerned about over-interpretation of the data and placing the finding in context with what is known of SCRIB membrane recruitment. I believe the authors have satisfied enough of the critiques to warrant publication. A few points which still need addressing:

"All three mutant constructs were efficiently recruited by TMIGD1 to cell-cell contacts (Fig. S3)" – There is still not enough evidence that experiments in 293 cells are reporting on "cell-cell contacts" rather than plasma membrane localization. In this experiment, it also appears that the bulk of PDZ1 and 2 mutants are not at the membrane.

Fig S5a – the imaging for TMIGD1 in the β -cat co-staining vs SCRIB co-staining looks very distinct (virtually all basolateral in the β -cat image). Is there an explanation for this?

The Δ D1 construct should be appropriately marked when used (eg S5a)

Fig 6 – while SCRIB P305L is not at the membrane in the Δ 5 expressing cells, it does appear completely co-localized with TMIGD1. Is there an explanation for this? What is the nature of the defect with the P305L mutant?

Reviewer #3 (Remarks to the Author):

The authors have thoroughly addressed my concerns, and this exciting manuscript on the membrane targeting of Scribble by TMIGD1 is ready for publication.

COMMSBIO-23-0672A

Membrane recruitment of the polarity protein Scribble by the cell adhesion receptor TMIGD1

Eva-Maria Thüring, Christian Hartmann, Janesha C. Maddumage, Airah Javorsky, Birgitta E. Michels, Volker Gerke, Lawrence Banks, Patrick O. Humbert, Marc Kvensakul, Klaus Ebnet

Point-by-Point Response to the Reviewer's Comments

Reviewer #1 (Remarks to the Author):

The authors have satisfactorily addressed most of my concerns.

A: NA

Reviewer #2 (Remarks to the Author):

The revised manuscript from Thüring et al. includes the following additional data:

- Fig 1c: reciprocal co-IP
- Fig 1d: quantitation of SCRIB levels following TAK243 addition
- Fig 6: SCRIB P305L localization with TMIGD1 fl/ Δ 5
- Fig 7b: cyst formation with TMIGD1 Δ 5
- Supp Fig 2: confocal images of HEK cells showing z stacks (β -CAT and SCRIB)
- Supp Fig 3: images in HEK cells of individual SCRIB PDZ mutants
- Supp Fig 4: SCRIB PDZ-Caax at Golgi (not ER)
- Supp Fig 5: SCRIB in MDCK cells described as "TMIGD1 recruits Scrib in polarized epithelial cells" along with MDCK expression of SCRIB LRR or PDZ

The authors have made an attempt to address the many concerns raised by the reviewers of the first submission. The major concerns were the weakness of the interaction in cells (compared to in vitro), the specificity of the interaction, a lack of clarity for the rationale of some experiments, the choice of 293 cells as a model for cell-cell adhesion and polarity, improved attempts to compare the interaction with other PDZ domain binders, and to derive more functional relevance from some of the experiments – particularly the cyst assay. Moreover, the reviewers were concerned about over-interpretation of the data and placing the finding in context with what is known of SCRIB membrane recruitment. I believe the authors have satisfied enough of the critiques to warrant publication. A few points which still need addressing:

"All three mutant constructs were efficiently recruited by TMIGD1 to cell-cell contacts (Fig. S3)" – There is still not enough evidence that experiments in 293 cells are reporting on "cell-cell contacts" rather than plasma membrane localization. In this experiment, it also appears that the bulk of PDZ1 and 2 mutants are not at the membrane.

A: As to the reviewer's point "*There is still not enough evidence that experiments in 293 cells are reporting on "cell-cell contacts" rather than plasma membrane localization.*" we disagree with the general notion that the evidence for a cell-cell contact localization of TMIGD1 is insufficient. We would like to focus the reviewer's attention to Suppl. Fig. S2 of our revised manuscript, in which we show confocal XZ-projections of ectopic TMIGD1 in HEK293 cells. These stainings show a very clear co-localization of TMIGD1/f.l. with β -catenin and Scrib, which both demarcate cell-cell junctions. In addition, the TMIGD1 signals as well as the β -catenin and Scrib signals are clearly absent from the

apical and basal membrane domains of these cells, which speaks against an indiscriminate membrane localization. Together with our previous observations in kidney-derived HK-2 cells (Hartmann et al (2020) BMC Mol Cell Biol. 21(1):30) and our stainings of ectopic TMIGD1/f.l. in HEK293 cells (Fig. 5 of this manuscript), we think that our notion on cell-cell contact localization of TMIGD1 and TMIGD1-mediated Scrib recruitment to cell-cell contacts is very strongly supported by our findings. We agree with the reviewer that the Scrib PDZ1 and PDZ2 mutants - in addition to their localization at the cell-cell contacts - are also localized in the cytoplasm. We consider several possibilities to explain this observation: First, a large number of Scrib-interacting proteins exist (Bonello and Peifer (2019) JCB 218:742) which may compete for Scrib binding. Mutating one of the Scrib PDZ domains may shift the balance in favor of a cytoplasmically localized Scrib interactor. Second, we ectopically expressed Scrib mutants. Ectopic expression does not reflect endogeneous expression levels, and a mislocalization of ectopically expressed proteins as a result of “protein overflow” is frequently observed. We would like to stress that the purpose of this experiment was to demonstrate functionality of the Scrib PDZ mutant constructs in membrane localization. We think that the results shown in Suppl. Fig. S3 demonstrate that all Scrib mutant constructs can localize to the membrane in a TMIGD1-dependent manner. As one additional note, we did not observe a comparable membrane localization in cells transfected with TMIGD1/ Δ 5 which is exclusively localized in the cytoplasm (Fig. 5, Suppl. Fig. S3A, for statistical evaluation see Suppl. Fig. S3B). In combination with the confocal stainings shown in Suppl. Fig. S2 we consider these data as strong evidence that TMIGD1 recruits Scrib to cell-cell contacts in HEK293 cells. We hope that we could clarify this point.

Fig S5a – the imaging for TMIGD1 in the β -cat co-staining vs SCRIB co-staining looks very distinct (virtually all basolateral in the β -cat image). Is there an explanation for this?

A: We do not see a principal difference in the localization of the Δ D1-TMIGD1 construct in the β -cat co-stainings vs Scrib co-stainings. In both cases, TMIGD1 is localized at the lateral membrane domain as indicated by the partial overlap of TMIGD1 with β -cat and Scrib. Also, the TMIGD1 signals localize along the entire lateral region indicating cell-cell contact localization. As to the difference in TMIGD1 localization at the apical membrane domain, we speculate that this might be due to different stages of brush border formation which is a highly dynamic process even in cultured cells (Crawley et al (2014) J Cell Biol. 207:441). We think that the stainings shown in Fig. S5A clearly support the view that TMIGD1 is localized at cell-cell contact sites in polarized epithelial cells. We hope that we could clarify the issue on the basolateral localization of TMIGD1.

The Δ D1 construct should be appropriately marked when used (eg S5a)

A: As requested, we have appropriately marked the Δ D1-TMIGD1 construct in Fig. S5A (other figures do not apply).

Fig 6 – while SCRIB P305L is not at the membrane in the Δ 5 expressing cells, it does appear completely co-localized with TMIGD1. Is there an explanation for this? What is the nature of the defect with the P305L mutant?

A: The P305L mutant of Scrib contains a Pro-to-Leu mutation at position 305 of Scrib. The P305 residue is conserved in vertebrates suggesting an important function. The function of this residue, however, is still unclear. It is possible that replacement of a Pro residue which acts as a helix breaker in secondary structures by a hydrophobic Leu residue impacts on the tertiary structure of the protein. As pointed out in our manuscript (Results section), Scrib/P305L does not localize at the membrane (Elsom and Humbert (2013) Cells Tissue Org. 198:1; Feigin et al (2014) Cancer Res. 74:3180). Expression of this mutant in mice promotes mammary gland tumorigenesis, and when expressed in MCF10A cells it

activates the PI3K signaling pathway and fails to suppress MAPK – Erk1/2 signaling (Elsum and Humbert (2013) *Cells Tissue Org.* 198:1; Feigin et al (2014) *Cancer Res.* 74:3180). Therefore, our observation of the robust recruitment of Scrib/P305L by TMIGD1 to the membrane provides a strong evidence for the physiological/pathophysiological relevance of the TMIGD1 – Scrib interaction. The observation that Scrib/P305L is not at the membrane in TMIGD1/ Δ 5-expressing cells provides additional evidence that the recruitment is most likely direct and is mediated by the PBM of TMIGD1. In summary, we think that the observations shown in Fig. 6 strongly support the main point of our manuscript, i.e. the recruitment of Scrib by TMIGD1 to the lateral membrane compartment which in simple epithelial sheets reflects cell-cell junctions. In the revised version of the manuscript we have rephrased the paragraph on the recruitment of the Scrib/P305L mutant by TMIGD1 to clarify this point (Results section).

Reviewer #3 (Remarks to the Author):

The authors have thoroughly addressed my concerns, and this exciting manuscript on the membrane targeting of Scribble by TMIGD1 is ready for publication.

A: NA